

# Mapping soil trace elements (Fe Mn Zn Ni) on the Tibetan Plateau
Huangyu Huo[1,2], Xiling Gu[1,2], Jiayi Li[1,2], Shanshan Yang[1], Yafeng Wang[1*], Jinzhi Ding[1*]
[1]State Key Laboratory of Tibetan Plateau Earth System, Environment and Resources (TPESER), Institute of Tibetan Plateau
Research, Chinese Academy of Sciences, Beijing 100101, China.
[2]University of Chinese Academy of Sciences, College of Resources and Environment, Beijing 100049, China.
*Corresponding to: Jinzhi Ding (jzding@itpcas.ac.cn), Yafeng Wang (yfwang@itpcas.ac.cn)
**Abstract.** Soil Micronutrients supply sustain critical ecological functions but exhibit poorly quantified distribution patterns in
high-altitude ecosystems. This study bridges this knowledge gap through a large-scale investigation across the Tibetan Plateau,
a cold-arid region where cryogenic weathering, aridity, and suppressed pedogenesis interact to govern microelement cycling.
We selected 526 spatially representative sites spanning climatic and edaphic gradients, analyzing six microelements (Fe, Mn,
Zn, Ni, Cu, Mo) alongside multi-factorial drivers (climate, vegetation, soil, topography, human disturbances, weathering
proxies). Random Forest modeling was employed to quantify controls and generate high-resolution spatial maps. Key results
reveal that pronounced regional heterogeneity driven primarily by moisture-related climatic variables (mean annual
precipitation, aridity index), with secondary modulation from weathering intensity and vegetation factors. Element-specific
spatial patterns were observed, with Fe enrichment in southeastern/southern plateaus, Mn gradients increasing southwestward
and Zn hotspots in central-eastern and western marginal zones. The machine-learning derived maps with a 1-km resolution
serve for benchmarking process-based microelement cycling models and rooting for sustainable ecosystem management
under climate change.



## 1 Introduction

As essential yet trace-level components of living systems, micronutrients (e.g., Fe, Mn, Cu, Zn, Ni, Mo) sustain fundamental ecological processes, including photosynthesis (Fe, Mn; Fischer et al., 2015; Schmidt et al., 2020), respiration (Fe; Dallman, 1986), enzymatic/redox functions (Cu, Zn, Mn; Hänsch et al., 2009), and biological nitrogen fixation (Ni, Mo; O'Hara, 2001). Crucially, micronutrient gradients in soils propagate through trophic chains, directly influencing human nutrition and health; deficiencies exacerbate global malnutrition burdens (Fageria et al., 2002; White et al., 2005). Despite their pivotal role in ecosystem stability and food security (Presteele et al., 2016; Stehfest et al., 2019), critical knowledge gaps persist regarding the distribution patterns and drivers of soil micronutrients from regional to global scales.

Soil micronutrient supply originates from coupled physicochemical weathering and biological mediation, critically regulated by local climate and topography (Ochoa-Hueso et al., 2020; Hartmann et al., 2023). In cold-arid high-altitude regions, particularly the Tibetan Plateau, extreme environmental interactions uniquely govern micronutrient cycling. Cryogenic processes such as glacial erosion and freeze-thaw cycles, accelerate physical bedrock weathering to mobilize lithogenic micronutrient reservoirs, while aridity concurrently constrains chemical weathering and elemental release (Mu et al., 2020; Mu et al., 2016). Low temperatures suppress biological turnover and synergize with aridity to compromise pedogenesis through clay deficits and diminished mineral reactive sites, thereby reducing elemental retention capacity (Dijkstra et al., 2004). These counteracting processes fundamentally shape microelements distribution patterns, yet remain severely understudied. Current research is largely restricted to localized transects (e.g., Heihe River Basin, Tibetan Plateau Highway) with limited spatial representation.

To address these knowledge gaps, we conducted a large-scale field investigation across the Tibetan Plateau, establishing 526 sampling sites distributed across representative temperature and moisture gradients (Fig. 1). The sampling design encompassed the plateau's dominant vegetation types and lithological classes. Using this dataset, we analyzed distribution patterns and key controlling factors for six essential trace elements (Fe, Mn, Cu, Zn, Ni, Mo,). We then applied a Random Forest algorithm to generate high-resolution spatial distribution maps of these microelements, representing the first comprehensive quantification at this scale and resolution.

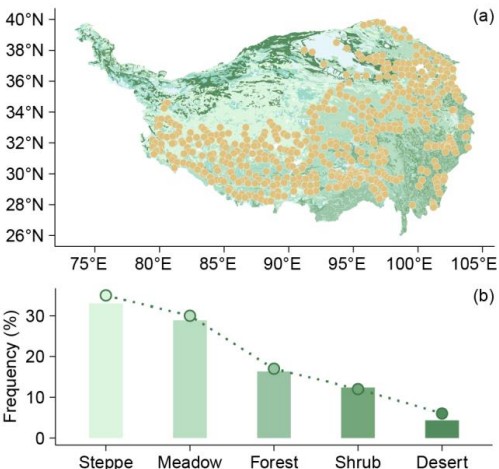


**Figure 1.** Sampling strategy and ecosystem representativeness. (a) Spatial distribution of sampling sites superimposed on China's
1:1,000,000 vegetation map. (b) Areal proportions of ecosystem types (bars) versus sampling point frequency distribution (dots) across
corresponding ecosystems.
**2 Methods**
**2.1 Field survey and soil microelements analysis**
We analyzed 1,660 topsoil samples collected from 526 locations during a 2019-2021 growing season (July-August) field
survey across the Tibetan Plateau (79-105°E, 27-40°N; Fig. 1a). Sampling sites represent major plateau ecosystems: forests,
shrubs, steppes, meadows, and deserts (Table 1). Site was selected using standardized criteria: maintaining relative
homogeneity in species composition, community structure, and habitat conditions, and avoiding proximity to roads or areas
with frequent human activity. At each location, we established a 15-m transect collecting triplicate soil samples (0-10 cm
depth) at 0 m, 7.5 m, and 15 m positions. Geographic coordinates, elevation, community type, and species composition
(Cheng et al., 2022) were systematically documented.
The portable Niton X-ray fluorescence (XRF) spectrometer was deployed in the field to determine total soil concentrations
of Fe, Mn, Cu, Zn, Ni, and Mo, leveraging its portability and compact design. Reference background values (Lindsay 1979)
include Fe: $3.8\times10^4$ mg kg$^{-1}$, Mn: $6.0\times10^2$ mg kg$^{-1}$, Cu: 30 mg kg$^{-1}$, Zn: 50 mg kg$^{-1}$, Ni: 40 mg kg$^{-1}$, and Mo: 1.7 mg kg$^{-1}$.
The contents of trace elements, including arsenic (As), barium (Ba), cadmium (Cd), cobalt (Co), chromium (Cr), lead (Pb),
strontium (Sr), and titanium (Ti), were determined using XRF spectroscopy. Powdered samples were pressed into pellets and
analyzed with a wavelength-dispersive XRF spectrometer. The instrument was calibrated using certified reference materials
to ensure analytical accuracy and comparability.





**Table 1.** Ecosystem classification with vegetation traits and sampling intensity.

| Biome | Characteristics | No. of samples | No. of locations |
|---|---|---|---|
| Steppe | Alpine steppes, dominated by cold-adapted herbaceous species such as Stipa purpurea, features sparse vegetation adapted to cold-arid conditions. | 578 | 183 |
| Meadow | Alpine meadows feature dense, low-stature vegetation sustained by year-round low temperatures, high humidity, and water-retentive soils. These ecosystems thrive on gentle slopes and valley floors at higher elevations, hosting relatively diverse flora with characteristic dominance of sedges including Kobresia pygmaea and K. humilis. | 505 | 156 |
| Forest | Forests on the Tibetan Plateau concentrate primarily in the southeastern region, dominated by high-altitude cold-temperate coniferous forests. These humid-adapted ecosystems feature fir (Abies) and spruce (Picea) species as characteristic components. | 280 | 92 |
| Shrub | Tibetan shrublands primarily occur in arid and alpine zones, characterized by low-growing, drought-tolerant dwarf shrubs such as Lonicera (honeysuckle) and Rhododendron species adapted to nutrient-poor soils and extreme climatic conditions. | 193 | 65 |
| Desert | Alpine deserts occur in extremely arid, cold regions and exhibit extremely sparse vegetation dominated by arid-tolerant dwarf shrubs and herbs. | 104 | 30 |

## 2.2 Soil Properties

Soil samples were sifted through 2 mm sieve, discarding visible stones and extracted roots. Soil pH was measured using the potentiometric method, and soil texture analysis, quantifying clay, silt, and sand content fractions, was determined using a laser diffraction particle size analyzer (Mastersizer 2000, Malvern, UK). The sieved samples were air-dried for elemental analysis. Soil organic carbon (SOC) content was quantified via the potassium dichromate oxidation method (Walkley-Black) with external heating. Total carbon (C) and total nitrogen (N) contents were measured using a Vario EL III elemental analyzer (Elementar, Germany). Total phosphorus (P) was extracted with sodium bicarbonate (Olsen method) and determined by molybdenum-antimony anti spectrophotometry. The concentrations of sulfur (S), potassium (K), calcium (Ca), sodium (Na), magnesium (Mg), and aluminum (Al) were determined using XRF spectroscopy. The chemical index of alteration (CIA) was calculated using the molar proportions of $Al_2O_3$, $CaO^*$, $Na_2O$, and $K_2O$ according to the formula: CIA = $Al_2O_3$ / ($Al_2O_3$ + $CaO^*$ + $Na_2O$ + $K_2O$). All oxide concentrations were determined by X-ray fluorescence spectroscopy and converted to molar units. $CaO^*$ represents CaO derived solely from silicate minerals, with carbonate contributions excluded where applicable.



### 2.3 Environmental variables

We considered geographic, climatic, biological, and edaphic drivers. Field measurements provided location (longitude, latitude), while slope and aspect data came from the National Tibetan Plateau Data Center (https://data.tpdc.ac.cn). The digital elevation model (DEM) data were collected from the Resource and Environment Science and Data Center (https://www. resdc.cn/). Climate variables, mean annual temperature (MAT) and mean annual precipitation (MAP) were downloaded from the Climate Data Store (https://cds.climate.copernicus.eu/#!/home). The Aridity Index (AI), calculated as mean annual precipitation/mean annual reference evapotranspiration, was obtained from the Global Aridity Index dataset (Trabucco et al., 2018), where higher values indicate greater humidity. Vegetation types (Forest, Shrub, Meadow, Steppe, Desert) followed the 1:1,000,000 China Vegetation Map classification (Hou, 2019). The normalized difference vegetation index (NDVI) data were obtained from an Earthdata Search (https://search. earthdata.nasa.gov/search). The net primary productivity (NPP) data were obtained from the study by Chen et al. (2023) and were calculated using the CASA model (Potter et al., 1993). The grazing activity data were obtained from statistical yearbooks. Based on the lithological data published by Dijkshoorn et al. in 2018, the rock types on the Tibetan Plateau were classified into acidic igneous rock (IA), acidic metamorphic rock (MA), clastic sedimentary rock (SC), carbonate rock (SO), aeolian facies rock (UE), and fluvial facies rock (UF).

### 2.4 Relative importance analysis and soil micronutrient mapping

Soil microelement measurements were preprocessed to detect and remove outliers exceeding the mean ± 3 standard deviations. To evaluate the relative importance of predictors in explaining soil micronutrient variability across the Tibetan Plateau, we applied the 'betasq' metric from the calc.relimp function in the R package *relaimpo* (Grömping, 2006), which is based on squared standardized regression coefficients and accounts for differences in variable scales and units. For spatial prediction, we developed six area-wide random forest models (each comprising 500 trees) targeting Fe, Mn, Cu, Zn, Ni, and Mo contents. The models were trained using a suite of environmental predictors, including topographic features (DEM, slope, aspect), climate variables (MAT, MAP, AI), vegetation indices (NDVI, NPP), soil properties (texture, SOC, pH, CIA), and anthropogenic disturbance (grazing intensity). Random forest was selected for its ability to model complex, nonlinear relationships and interactions among diverse types of predictors. Model hyperparameters were optimized using grid search combined with tenfold cross-validation. To assess model generalizability, we examined the extent to which the predictor parameter space in the validation set overlapped with that of the original training data. Model performance was evaluated by comparing predicted versus observed values using scatterplots (predicted on the x-axis, observed on the y-axis) following the

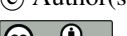

method of Piñeiro et al. (2008), with models achieving strong predictive performance ($R^2 = 0.6–0.7$). All statistical analyses
were conducted using R version 3.4.4.
**3 Results**
**3.1 Soil elements ranking**
As shown in Figure 2, Ca and C had the highest mean concentrations, reaching 30,462.47 mg·kg⁻¹ and 30,361.14 mg·kg⁻¹,
respectively. Their relatively large standard deviations indicate substantial variability across sampling sites. K and Al also
showed relatively high mean concentrations, at 16,788.79 mg·kg⁻¹ and 14,166.30 mg·kg⁻¹, respectively. A secondary tier
included Mg, N, Na, S, and P. Soil Fe content approached those of macroelements like Ca and C. Mn shares a similar order
of magnitude with Na and S, while Cu and Zn demonstrated comparable mean concentrations. Mo consistently registered the
lowest content among all measured soil elements.
Frequency distribution analysis showed that soil Fe, Mn, Zn exhibited near-normal distributions, evidenced by closely
aligned median and mean values (Fig. 2). In contrast, Cu, Ni, and Mo showed right-skewed distributions, with most samples
clustering at lower contents. Fe spanned a broad concentration range (3,339.62-54,877.54 mg kg⁻¹), highlighting its
abundance and widespread spatial distribution in soils. Mn also displayed substantial spatial variation (mean: 576.74 $\pm$
206.44 mgkg⁻¹; CV: 35.8%), while Cu, Zn, Ni, and Mo exhibited notably lower mean contents (25.32 $\pm$ 9.28, 27.24 $\pm$
8.55, 49.35 $\pm$ 14.03, and 4.63 $\pm$ 1.14 mg kg⁻¹, respectively).

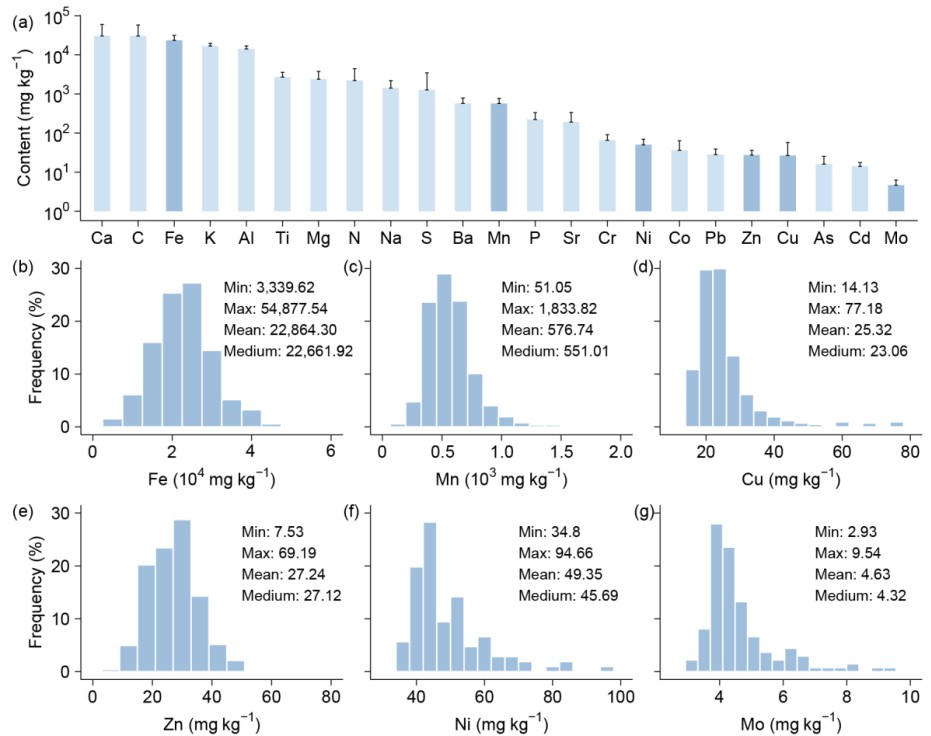

**Figure 2.** Content hierarchy and frequency distributions of soil microelements (Fe, Mn, Cu, Zn, Ni, Mo) across the Tibetan Plateau. (a) Elemental ranking by mean content, (b-g) Frequency distribution histograms for each microelement showing spatial heterogeneity patterns.

## 3.2 Soil microelements across Vegetation types

Soil microelements contents (Fe, Mn, Cu, Zn, Ni, and Mo) varied significantly among different vegetation types (Fig. 3). Fe contents were highest in shrub and forest ecosystems, with mean values of 26,264.11 mg·kg⁻¹ and 26,090.66 mg·kg⁻¹, respectively, exceeding values observed in desert (19,762.66 mg·kg⁻¹) and steppe ecosystems (19,852.37 mg·kg⁻¹) by 31-33%. Similarly, Mn contents were greatest in forest (703.22 mg·kg⁻¹), followed by shrub (606.33 mg·kg⁻¹) and meadow (591.59 mg·kg⁻¹), while the lowest Mn levels were recorded in desert and steppe ecosystems (both below 510 mg·kg⁻¹).

Soil Cu showed minimal variation across vegetation types, with mean values ranging narrowly between 25.23 and 25.91 mg·kg⁻¹. Forest soils had slightly higher Cu levels, but the differences were not statistically significant, suggesting a relatively uniform spatial distribution of Cu across ecosystems. Zn, however, demonstrated a strong vegetation-dependent variability, with the highest mean contents found in forest (32.00 mg·kg⁻¹) and shrub (31.15 mg·kg⁻¹) ecosystems. In contrast, Zn levels were markedly lower in steppe (22.94 mg·kg⁻¹) and desert (21.95 mg·kg⁻¹) soils, with differences exceeding 10 mg·kg⁻¹, indicating pronounced biogeochemical variation.

In contrast to the previous elements, Ni and Mo exhibited distinct patterns across vegetation types, with notably lower
contents in forest and shrub ecosystems. Ni contents were highest in meadow soils (54.34 mg·kg⁻¹), followed by shrub
(52.66 mg·kg⁻¹), steppe (48.78 mg·kg⁻¹), desert (43.87 mg·kg⁻¹), and forest ecosystems (41.73 mg·kg⁻¹). Soil Mo contents
showed negligible differences among ecosystems, though steppe exhibited marginally higher levels. No statistically
significant differences were observed between vegetation types, indicating minimal vegetation control over Mo distribution.

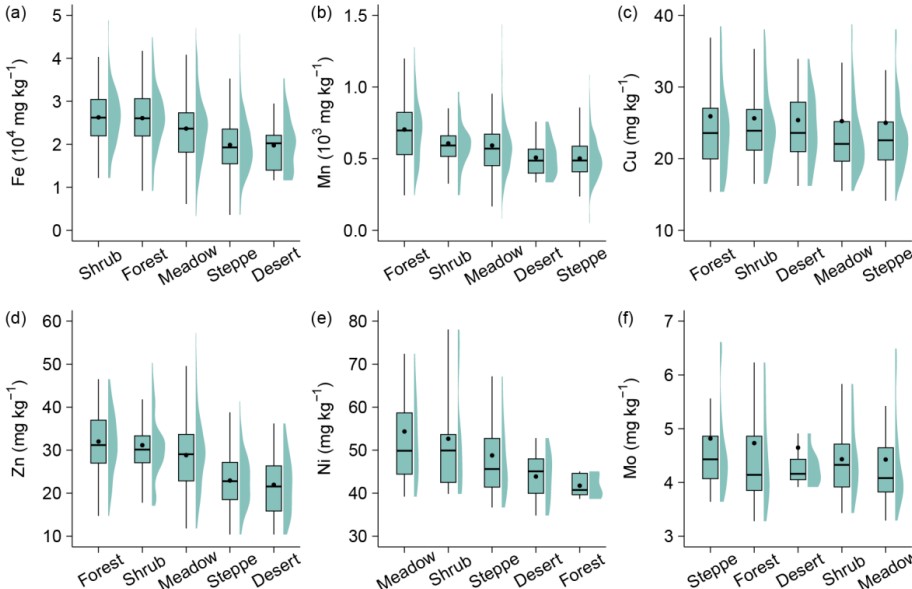

**Figure 3.** Variability in soil microelement contents (Fe, Mn, Cu, Zn, Ni, Mo) across Tibetan vegetation types. Boxplots show the data
distributions for each vegetation type. Within each plot, the boxes represent the interquartile range (IQR), the horizontal lines within boxes
indicate the median values, and black dots denote the mean values. The whiskers extend to 1.5 times the IQR. The surrounding shaded
violin shapes indicate the kernel density distribution of the data.
**3.3 Soil microelements across lithological classes**
Lithological class exerted a significant influence on the spatial distribution of certain soil microelements (Fig. 4). Fe contents
differed notably across lithological classes, with the highest values observed in soils derived from acidic metamorphic rocks
(mean value: 25,252.72 mg·kg⁻¹), followed by those from carbonate rocks (24,260.96 mg·kg⁻¹) and eolian facies rocks
(23,902.77 mg·kg⁻¹). The lowest Fe contents were recorded in soils developed from acidic igneous rocks (20,830.15
mg·kg⁻¹). A similar geochemical pattern pattern was observed for Mn, with the maximum contents in
acidic-metamorphic-drived soils (658.37 mg·kg⁻¹), and the lowest in acidic-igneous-drived soils (530.45 mg·kg⁻¹).
Zn contents in soils also varied across lithologies, with the highest values associated with acidic metamorphic rocks (mean
value of 30.79 mg·kg⁻¹), followed by eolian facies rocks and clastic sedimentary rocks, while relatively lower levels were
found under acidic igneous rocks (25.27 mg·kg⁻¹) and fluvial facies rocks (25.10 mg·kg⁻¹). In contrast, Cu and Mo showed
no significant differences among lithological classes. Cu contents were slightly higher under fluvial facies rocks (25.60
mg·kg⁻¹), but overall differences were minimal. Similarly, Mo contents were relatively uniform, with slightly elevated levels
under acidic metamorphic rocks and eolian facies rocks (both ~4.86 mg·kg⁻¹), and lower values under clastic sedimentary
rocks and carbonate rocks (both <4.48 mg·kg⁻¹), though differences were not statistically significant.

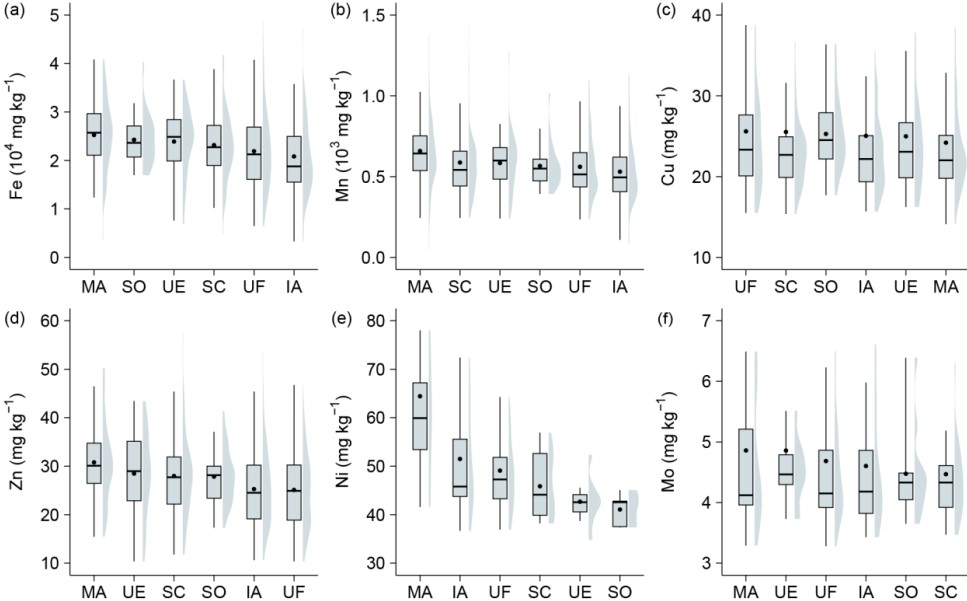


**Figure 4.** Variability in soil microelement contents (Fe, Mn, Cu, Zn, Ni, Mo) across Tibetan lithological classes. Boxplots show the data
distributions for each lithological classes. Within each plot, the boxes represent the interquartile range (IQR), the horizontal lines within
boxes indicate the median values, and black dots denote the mean values. The whiskers extend to 1.5 times the IQR. The surrounding
shaded violin shapes indicate the kernel density distribution of the data. Abbreviations of lithological classes: IA = acidic igneous rock,
MA = acidic metamorphic rock, SC = clastic sedimentary rock, SO = carbonate rock, UE = eolian facies rock, UF = fluvial facies rock.

Soil Ni contents showed the most pronounced variation among lithologies, with the highest level recorded under acidic
metamorphic rocks (64.43 mg·kg⁻¹), followed by acidic igneous rocks (51.49 mg·kg⁻¹). Soils developed from clastic
sedimentary rocks and carbonate rocks had significantly lower Ni contents (45.85 and 41.06 mg·kg⁻¹, respectively). In
summary, Fe, Mn, and Zn were substantially enriched in soils derived from acidic metamorphic rocks. Cu and Mo showed
relatively uniform distributions across lithologies Ni was notably elevated under acidic metamorphic rocks.





### 3.4 Drivers of soil micronutrient pattern

Relative importance analysis considered five variable groups, including climate, vegetation, soil properties, topography, and human disturbances. Among all investigated variables, climatic factors dominantly control soil micronutrient (Fe, Mn, Cu, Zn, Ni) distribution across the Tibetan Plateau (Figs. 5a-f). Regional moisture conditions, characterized by mean annual precipitation (MAP) and aridity index (AI), were the primary drivers. MAP consistently ranked as the top predictor for Fe, Mn, Zn, and Ni. Vegetation indicators (e.g., NDVI, NPP) also showed high importance for Mo and Zn. Soil properties (pH, SOC, texture) and topography (slope, aspect, elevation) contributed to distribution patterns but exhibited lower relative importance.

The distribution of Fe was primarily regulated by climatic conditions (especially precipitation) and parent material weathering intensity (represented by the chemical index of alteration, CIA), with secondary contributions from normalized difference vegetation index (NDVI), aridity index (AI), net primary productivity (NPP). For Mn distribution, Climate (MAP, AI) and soil properties (CIA, soil texture) were dominant. Soil Zn also show highly sensitive to climate (MAP, AI), weathering intensity (CIA), and vegetation cover (NDVI). Ni distribution was predominantly controlled by natural environmental conditions including MAP, AI, MAT and topography. For both Cu and Mo, climate variables (AI, MAP) and vegetation indicators (NPP or NDVI) consistently ranked among the top three factors governing their spatial distribution.

The partial dependence plots revealed distinct responses of soil microelement to key environmental drivers (Fig. 6). Five elements (Fe, Mn, Cu, Zn, Ni) exhibited a typical U-shaped relationship with mean annual precipitation, showing higher contents in both low and high precipitation zones, and a clear trough in the intermediate range (approximately 300–500 millimeters). This pattern aligned closely with responses to drought indices, confirming shared moisture sensitivity. Increases in the chemical index of alteration were generally associated with elevated levels of Mn, Ni, and Cu, particularly when the chemical index of alteration exceeded 0.5. In summary, the spatially heterogeneous distribution of Tibetan soil microelements is co-regulated by precipitation, vegetation, and chemical weathering intensity.

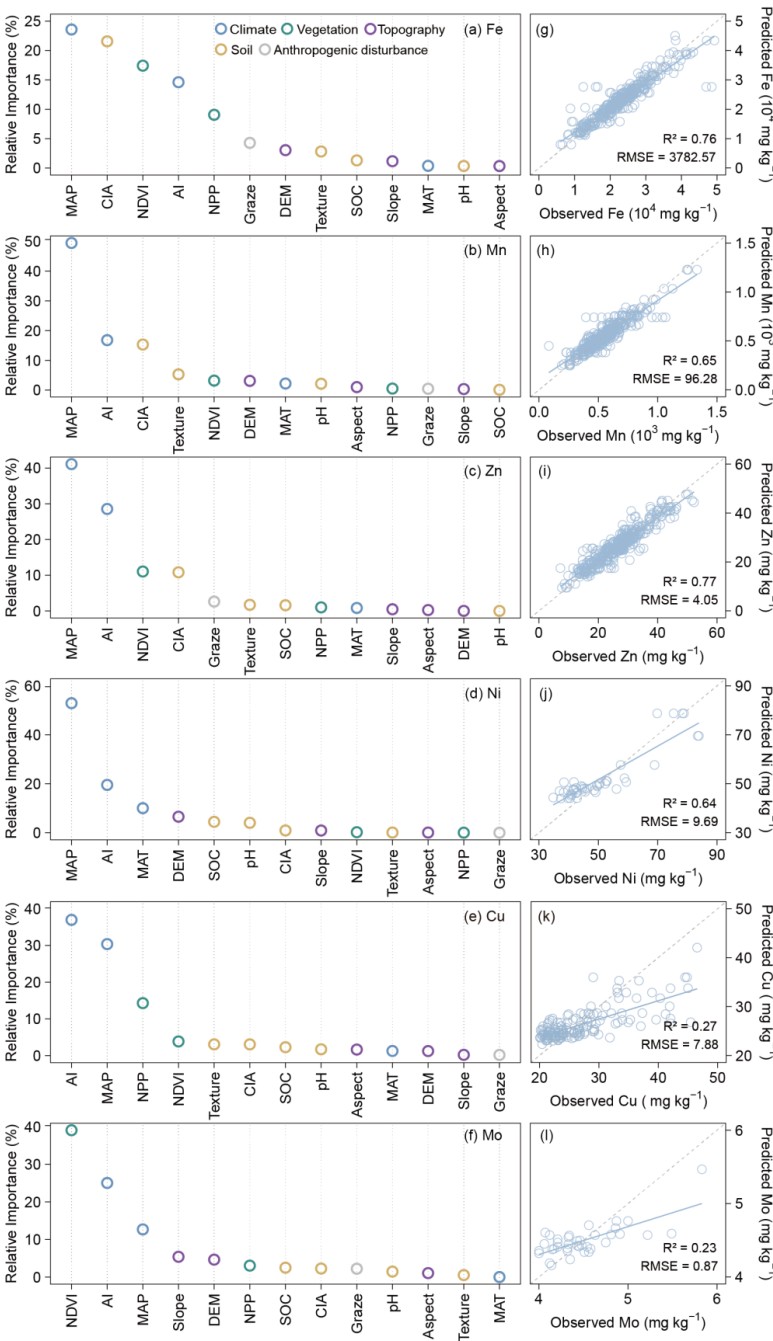

**Figure 5.** Relative importance of biotic and abiotic factors for soil microelements (Fe, Mn, Zn, Ni, Cu, Mo) on the Tibetan Plateau (a-f).
Relationship between observed and predicted values of soil micronutrients (Fe, Mn, Zn, Ni, Cu, Mo) on the Tibetan Plateau based on the
Random Forest model (g-l). The blue solid line represents the fitted relationship using ordinary least squares regression, while the gray
dashed line indicates the 1:1 line between observed and predicted values.

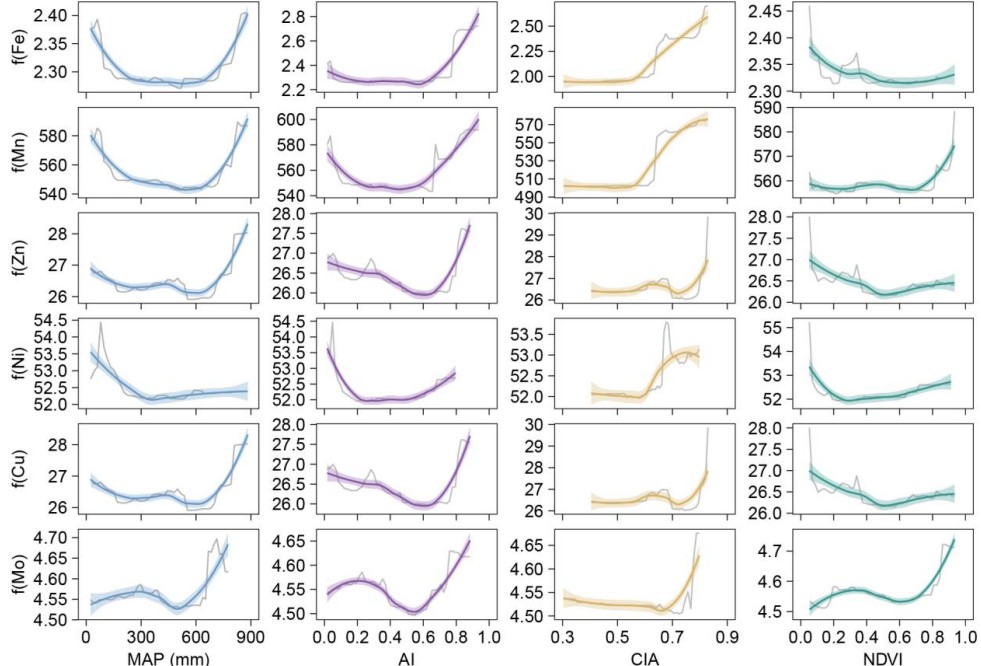

**Figure 6.** Partial dependence of six soil microelements (Fe, Mn, Zn, Ni, Cu, Mo) on four predictive variables: mean annual precipitation (MAP), aridity index (AI), chemical index of alteration (CIA), and normalized difference vegetation index (NDVI). Gray lines represent the original partial dependence, while smoothed fits for each element are shown as colored lines. Shaded areas represent the 95% confidence intervals.

**3.5 Soil micronutrients maps**

We employed random forest modeling to predict the spatial distribution of six soil micronutrients across the Tibetan Plateau. Model performance varied among elements (Figs. 5g–l). The model achieved the highest predictive accuracy for Zn and Fe, with $R^2$ values of 0.77 and 0.76, respectively (Figs. 5g and 5i), indicating that the spatial variability of Zn and Fe is well captured by the selected environmental predictors. Mn and Ni models also showed moderate performance, with $R^2$ values of 0.65 and 0.64 and corresponding RMSEs of 96.28 and 9.69 (Figs. 5h and 5j). In contrast, the models for Cu and Mo displayed poor predictability, with $R^2$ values of only 0.27 and 0.23 (Figs. 5k and 5l). Overall, the model evaluation results suggest that the random forest approach is effective in predicting the distributions of Fe and Zn, moderately reliable for Mn and Ni, and limited utility for Cu and Mo.

**Figure 7** illustrates the spatial patterns of soil microelements (Fe, Mn, Zn, Ni) across the Tibetan Plateau, as predicted by random forest models. The resulting maps reveal significant spatial heterogeneity of these elements. The highest contents of Fe are primarily located in the southeastern region, the southern margins, and parts of the western plateau. Mn shows a

distinct gradient, with contents increasing from northeast to southwest. Predicted Mn values range from 298.47 to 1,110.19
mg·kg⁻¹, and the areas with the highest Mn contents are mainly distributed in the humid southeastern and southern parts of
the plateau. Zn displays relatively high contents in the central-eastern region and along certain western edges of the plateau,
with predicted values ranging from 17.09 to 45.03 mg·kg⁻¹. Ni has a narrower predicted content range (44.28–61.95 mg·kg⁻¹)
and shows a more spatially homogeneous distribution. However, localized hotspots of elevated Ni contents are observed in
parts of the northeastern and southern plateau.

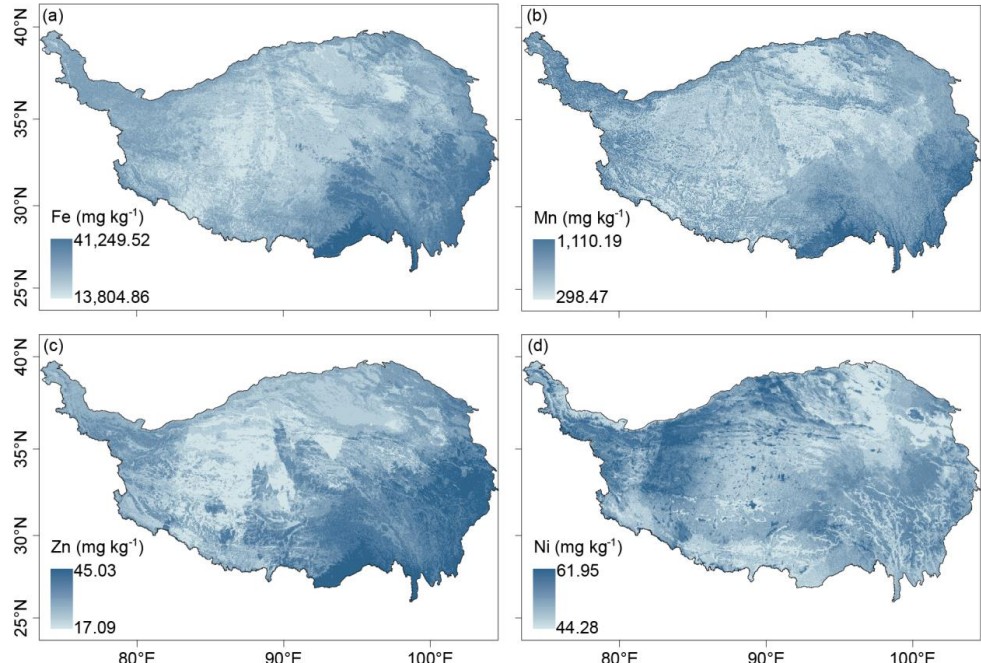


**Figure 7.** Spatial distribution of soil microelements (Fe, Mn, Zn, Ni) on the Tibetan Plateau.
**4 Discussion**
Our results indicate generally low contents of soil micronutrients (Fe, Mn, Cu, Zn, Ni, Mo) across Tibetan Plateau
ecosystems (Fig. 2), consistently falling below well-established global averages for reference soils (Lindsay 1979). These
deficient levels carry profound long-term ecological implications, including the risk of irreversible depletion of lithogenic
microelement pools (Jones et al. 2013), given their replenishment cycles operate on geological timescales (million years).
Furthermore, accelerated warming may exacerbate microelement dilution effects, thereby increasing regional soil
degradation vulnerability (Clair & Lynch, 2010; Myers et al., 2014; Pachauri et al., 2014).



The distribution of soil microelements across the Tibetan Plateau demonstrates significant spatial heterogeneity (Figs. 2 and
7), aligning with patterns observed in Europe for elements such as Zn. This variability is predominantly governed by the
interactions among climate, vegetation, and soil (Figs. 5a-f and 6). Notably, precipitation emerges as the primary predictor
for elements such as Fe, Mn, Zn, and Ni, with all except Mo exhibiting characteristic U-shaped responses, with minima
occurring at 300-500 mm. This pattern likely reflects distinct weathering regimes across precipitation gradients. In arid
regions (<300 mm), physical weathering processes, including freeze-thaw fracturing and aeolian erosion, predominate,
allowing trace elements (e.g., Ni, Mo) to accumulate near the surface due to evaporation effects (Pachauri et al., 2014;
Moreno-Jimenez et al., 2023). In transitional precipitation regimes (300-500 mm), intensified chemical weathering occurs;
however, leaching fluxes surpass the rates of parent material weathering, leading to soil elemental depletion (Anderson, 2019;
Bluth et al., 1994; Hartmann et al., 2014). In humid regions (>500 mm), enhanced chemical weathering results in the
formation of secondary clay minerals (e.g., montmorillonite, illite), whose negatively charged surfaces facilitate elemental
retention through ionic adsorption and co-precipitation mechanisms (Alloway, 2009).
Our findings suggest that the aridity index is a significant determinant of soil microelement distribution. Specifically,
elemental contents tend to decrease when the aridity index falls below a certain threshold. This trend likely reflects reduced
input or retention of elements under arid conditions. Drought conditions may modify soil redox states, thereby influencing
element speciation, adsorption capacity, mobility, and ultimately, leaching behavior (Brady et al., 2016; Loveland et al., 2003;
Carter et al., 1995). Also, arid environments may indirectly impact trace elements through alterations in soil pH and soil
organic matter content (Moreno-Jimenez et al., 2019). Previous research has shown that droughts induced by climate change
can restrict the availability of essential microelements, such as iron and zinc. This limitation, along with other adverse effects
like diminished water availability, poses substantial threats to vital ecological processes and services in drylands, including
food production (Gupta et al., 2008; Graham, 1991).
Our random forest regression models demonstrated robust predictive capability (e.g. cross-validated $R^2$ range from 0.64 to
0.77 for Fe Mn Zn Ni). Nevertheless, model accuracy could be further improved to more extensive field sampling and
refinement of input data. Specifically, targeted collection of soil microelement data and associated covariates in
underrepresented high-altitude regions of the Tibetan Plateau is necessary to address existing spatial gaps. Additionally,
systematic reduction of uncertainties inherent in gridded environmental datasets is essential, as these uncertainties propagate
errors into microelement predictions. Continued advancement in both field observations and foundational geospatial dataset
is crucial for improving the reliability of regional-scale element mapping.



**5 Data availability**
The gridded soil trace element (Fe Mn Zn Ni) maps for Tibetan Plateau can be downloaded from https://doi.org/
10.11888/Terre.tpdc.302870 (Huo et al., 2025).
**6 Conclusions**
This study delivers a comprehensive assessment of spatial distribution patterns for six soil micronutrients (Fe, Mn, Cu, Zn,
Ni, Mo) across the Tibetan Plateau, revealing pronounced regional-scale heterogeneity. Moisture-related variables (e.g.,
mean annual precipitation, aridity index) are the primary drivers of microelement distributions, with significant secondary
modulation by weathering intensity and vegetation factors. These findings highlight the coupled effects of climate,
vegetation, and parent material on microelement biogeochemical cycling within the complex environmental context of the
Tibetan Plateau. Using five predictor groups (climate, vegetation, soil properties, topography, and human disturbances), we
generated high-resolution spatial maps for four well-predicted elements (Fe, Mn, Zn, Ni) via machine learning. These maps
provide validated initial conditions for process-based models simulating microelement cycling, advances understanding of
elemental distribution in alpine ecosystems.
**Author contributions.**
JZ conceived the study. HY conducted the field survey and was responsible for data collection and processing. HY prepared
the manuscript with contributions from all co-authors.
**Competing interests.**
The contact author has declared that none of the authors has any competing interests.
**Acknowledgements.**
This study was supported by the Second Tibetan Plateau Scientific Expedition and Research Program (2022QZKK0101),
National Natural Science Foundation of China (42471159), and Chinese Academy of Sciences (CAS) Project for Young
Scientists in Basic Research (YSBR-037).







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
