# Peer review of "Mapping soil trace elements (Fe Mn Zn Ni) on the Tibetan Plateau"

_Earth System Science Data, 2025_

## Author Comment (AC1)

**Response to Reviewer #1**

The manuscript submitted by Huo et al. provided a dataset of six soil trace elements (I think they tried to focus on micronutrients) across the Tibetan Plateau (TP). The element distribution was investigated, and the possible factors regulating their distribution were discussed. Meanwhile, they used AI models to predict and map these elements across the TP. In general, this dataset is important to understand micronutrient cycling in the third pole of the world. However, there are many big issues limiting its wide use.

[Response] We sincerely thank the reviewer for the insightful and constructive comments on our manuscript. In response, we have carefully revised the manuscript and provided detailed point-by-point replies to each suggestion. These revisions have greatly improved the quality and scientific rigor of the work, and we hope that the revised version meets the reviewer's expectations.

[Comment 1] The first issue is the analysis methods of the elements in the soil, which used the XRF to determine the element concentrations in the field. I think this method has a large uncertainty when using in the field, compared to traditional methods like ICP-OES and ICP-MS. Unfortunately, the authors did not give convincing quality control data to ensure the precision of the analysis.

[Response] Many thanks for raising this important concern. In our study, we initially used a second-generation portable XRF to conduct in situ measurements (n=130), but because this device could not reliably detect Mo, a key micronutrient of our focus, all samples were re-analyzed in the laboratory using a third-generation XRF with improved detection accuracy. All data reported in this study for core micronutrients (Fe, Mn, Cu, Zn, Ni, Mo, V) were obtained from laboratory-based wavelength-dispersive XRF (third-generation XRF) on air-dried, and sieved (<2 mm) samples.

To further validate the data quality, 218 randomly selected samples were re-analyzed using reviewer mentioned ICP-MS, a widely accepted traditional methods. Strong correlations were observed between XRF and ICP-MS for Fe, Mn, Cu, V ( $R^2 = 0.7$ -0.9; Figure R1), closely aligning with the 1:1 line. For Zn and Ni, significant correlations were also found ( $R^2 = 0.4$ -0.5), though with systematic deviations from the 1:1 line. Taking ICP-MS as the benchmark, XRF slightly over- or

underestimated absolute contents. Importantly, these deviations do not substantially affect spatial distribution patterns central to this study for the following reasons: First, the XRF-based observed values are highly correlated with the XRF-based predicted values, and the ICP-MS-based observed values are also highly correlated with the ICP-MS-based predicted values (R2 = 0.85-0.95; **Figure R2**). Second, we calibrated XRF-based Zn, and Ni using the regression relationships between the two methods and repeated spatial mapping. Results show that the spatial distribution patterns before and after calibration were highly consistent (R2=0.8-0.9; **Figure R3**), confirming that the main conclusions are not sensitive to these measurement uncertainties. Third, for Mo, the results obtained from the two methods were not fully consistent, likely due to its low content. The XRF-based measurements may involve considerable uncertainties (**Figure R4**). Therefore, we have included both the observational and predicted results in the supplementary materials and discussed their potential uncertainties in detail in the Discussion section.

**We have added data quality control details in the Methods section as follows:**

"To ensure the reliability of soil micronutrient measurements, we adopted a two-step analytical strategy. First, in situ measurements were conducted using a second-generation portable Niton X-ray fluorescence (XRF) analyzer to obtain preliminary contents under natural field conditions. However, because this instrument could not reliably detect Mo, all soil samples were subsequently air-dried, sieved (2 mm), and re-analyzed in the laboratory using a third-generation XRF. Compared with the second-generation device, the third-generation XRF offers an extended detection range and improved accuracy (Lemière, 2018). All data presented in this study are based on the laboratory XRF measurements".

To further validate the laboratory XRF results, a subset of 218 samples was randomly selected and re-analyzed using ICP-MS, a widely accepted reference method (Simon, 2005). The two methods showed strong correlations for Fe, Mn, V and Cu ( $R^2 = 0.67-0.89$ , P

**Figure R1.** Comparison between laboratory-based third-generation XRF and ICP-MS measurements for six soil micronutrients (Fe, Mn, Cu, Zn, Ni, V).

**Figure R2.** XRF-based observed values are highly correlated with the XRF-based predicted values (a-b), and the ICP-MS-based observed values are also highly correlated with the ICP-MS-based predicted values (c-d).

**Figure R3.** Spatial correlation maps of Zn and Ni contents before and after calibration of XRF measurements against ICP-MS. The strong consistency confirming that the calibration adjusted absolute contents but did not alter the large-scale spatial patterns of these elements.

**Figure R4.** Comparison between laboratory-based third-generation XRF and ICP-MS measurements for Mo. The results obtained from the two methods were not fully consistent, likely due to its low content. The XRF-based measurements may involve considerable uncertainties. Therefore, we have included both the observational and predicted results in the supplementary materials and discussed their potential uncertainties in detail in the Discussion section.

[Comment 2] Second, I do not why only six elements were considered in this study. There are many trace elements or even micronutrients in soil, and some others (e.g., B, V) are also important for plants or animals. These limited element types in this study are not helpful for users to apply further studies.

[Response] In this study, we mainly focused on key micronutrients because of their critical roles in plant nutrition and biogeochemical cycling. As the reviewer mentioned, B is indeed an essential nutrient, but it was not measured in this study due to methodological limitations. For V, we followed the reviewer's suggestion and added the results into the main text (Figure R5).

**Figure R5.** Observed statistics and predicted spatial map of V contents across Tibetan Plateau soils.

[Comment 3] Meanwhile, there is many key background information (specific elevation, vegetation type in each site, local climate) that was not provided in the manuscript. Such data are also the important part of the dataset.

[Response] Following your suggestion, more background information, including site-specific elevation, soil properties, vegetation type, and local climate conditions, has been compiled and provided in the Supplementary Table in the revised manuscript.

[Comment 4] Third, the data predicted, as the authors mentioned in the manuscript, also have many uncertainties. One of the reasons may be linked to selection of ecosystem types. In this study, many natural ecosystems were selected, but farmlands (a landscape associated with human disturbance) were ignored. This should result in the uncertainty of these element distributions on the TP.

**[Response]** On the Tibetan Plateau, cropland area is relatively small, accounting for less than 2% of the region (**Figure R6**) according to the 1:1,000,000 Vegetation Map of China. In our original manuscript, cropland samples (n = 12, ~2% of total) were merged into the grassland category due to their small area.

In the revised manuscript, we treated cropland as an independent ecosystem type in our spatial comparison and up-scaling analysis (**Figure R7**). This revision better captures the influence of human disturbance while providing a more accurate representation of micronutrient distributions across different ecosystem types on the Plateau.

**Figure R6.** Areal proportions of ecosystem types (bars) versus sampling point frequency distribution (dots) across corresponding ecosystems. Similar bar and dot heights indicate that the sampling is proportionally representative.

**Figure R7.** Soil micronutrient contents across six ecosystem types (forest, shrub, meadow, steppe, desert, and cropland).

[Comment 5] Additionally, the section of Discussion was not well drafted, which is very superficial and lacks key evidence to support the discussion points.

[Response] In the revised manuscript, we have substantially strengthened the Discussion section to provide deeper and more evidence-based interpretations:

First, we included comparisons of our results with the few available observations on the Tibetan Plateau, as well as broader datasets from other regions, to better place our findings in a wider context (**Table R2**). "In terms of data range and mean values, the contents and magnitudes of the soil micronutrients we observed are generally consistent with previous reports from the Plateau and global grassland soils, indicating the reliability of our datasets. For the Plateau, the mean Zn, and V contents

in this study were slightly lower, while Ni were somewhat higher (**Figure R8**). These discrepancies are more likely due to differences in study regions. Our sampling covered a much broader area, whereas previous studies focused mainly on local regions such as the Heihe Basin (Bu et al., 2016). Given the substantial spatial heterogeneity of soil micronutrients across the Tibetan Plateau, such differences are expected and further highlight the necessity of exploring soil micronutrient patterns at the plateau scale" (see our reply to the comment 34).

Second, we performed a more rigorous statistical analysis using accumulated local effects (ALE) and partial dependence plots (PDP) to characterize the nonlinear relationships between soil micronutrients and MAP. Both ALE and PDP revealed consistent trends, confirming the robustness of the nonlinear precipitation effects on micronutrient distributions. We also examined relationships among MAP, CIA, and soil micronutrients, and found a significant positive correlation that may help explain the patterns. However, our observations alone cannot fully account for the U-shaped micronutrient response, so we treat this U-shaped pattern as an open question for future investigation (see our reply to the comment 37).

Third, we added a detailed explanation of the uncertainties in both our observed and predicted datasets. On one hand, we provided an in-depth discussion of the methodological uncertainties associated with XRF measurements and data quality control (see Response to Comment 1). On the other hand, we expanded the discussion on the uncertainties inherent in the machine-learning upscaling approach: "Our random forest regression models demonstrated robust predictive capability (e.g. cross-validated R2 range from 0.64 to 0.77 for Fe Mn Zn Ni). Cu and Mo exhibit weaker predictive performance, which may be attributed to the following reasons: First, analytical limitations may introduce measurement error, as both elements occur at very low contents, especially Mo. Second, key process controls are not fully captured by the current predictors. E.g. Cu is strongly influenced by nonlinear interactions with Fe/Al oxides, whereas Mo is affected by carbonate content, redox conditions (Tack et al., 1995). These mineralogical and redox variables are underrepresented, leading to potential bias. In addition, both elements show hotspot-prone, right-skewed distributions (Figure 2), likely due to localized anthropogenic sources". We have added these points into revised Discussion. Also, we added pixel-level uncertainty layers with the maps for users to properly utilize these datasets (also see our reply to the comment 4, #2).

Also, we clarified the scope of applicability of the predicted maps: "We note that ecosystem-specific pedogenesis (e.g., organic-rich surface layers in some forests vs. thin mineral A horizons in grasslands) can contribute to differences in surface micronutrient contents. Consequently, part of the spatial variation we report likely reflects vertical horizon contrasts sampled at a uniform 0-10 cm depth. Users should consider this context when applying the dataset, especially for process inference or when comparing across ecosystems with contrasting surface horizons. Where applications require deeper profiles or horizon-specific interpretation, we recommend integrating our maps with local profile data" (also see our reply to the comment 19).

We removed overstated or speculative conclusions from the original version and supported the revised discussion with direct evidence from our dataset. Together, these revisions have improved the depth, rigor, and clarity of the Discussion section.

**Figure R8.** Comparison of mean soil micronutrient contents (Fe, Mn, Cu, Zn, Ni, and V) between this study and previous studies (Yang et al., 2020; Sheng et al., 2012; Cheng et al., 1993) on the Tibetan Plateau. Bars represent mean values, and error bars indicate one standard deviation.

[Comment 6] Line 1: Normally, Fe cannot be termed as a trace element in soil (but in plants, it can be termed as micronutrient), and it has a high abundance in crust or soil like K, Ca, and Mg.

[Response] We agree with this correction. Our intention was to emphasize elements that function as micronutrients for plants rather than to classify them strictly as trace elements in soils. We have revised the manuscript by replacing "trace elements" with "micronutrients" to avoid confusion.

[Comment 7] Furthermore, in Lines 10-11, six elements are targeted in this study, but why only four of them is shown in the title? Even though there are six elements targeted, I think the dataset is still small. There are many kinds of trace elements in soil, such as toxic metals (e.g., Cd, Cr, Hg, Pb, Sb) and micronutrients (e.g., V, B). So, I strongly suggest the authors to adding more elements in the dataset. This will increase the application of the dataset and attract more attention.

[Response] As stated in the manuscript, this study mainly focused on key micronutrients for plants. Following the reviewer's suggestion, we have added vanadium (V) to the dataset and provided the results in the main text. Boron (B), although important, was not measured in our study. Other toxic metals (e.g., Cd, Cr, Hg, Pb, Sb) were not included because they fall outside the scope of this study, which emphasizes nutrient elements rather than contaminants. We appreciate the reviewer's understanding of this limitation.

In the original manuscript, only four elements were highlighted in the title because the spatial prediction models for Mo and Cu performed less robustly, as the available predictors did not fully capture their spatial variability. Therefore, these two elements were not extrapolated to the Plateau scale in the previous version. In the revised manuscript, we have conducted spatial predictions for all six elements, and their associated uncertainties are explicitly clarified in the Discussion section. The title has also been updated as "Mapping key soil micronutrients across the Tibetan Plateau".

[Comment 8] Line 7: Micronutrients are totally different from trace elements (shown in the Title). Micronutrient is defined by plant demand, but trace element has a broader scope. As I mentioned above, some toxic elements cannot be termed as micronutrients, but they belong to trace elements. What does this dataset target to, micronutrients or trace elements? If you aimed to map micronutrients, the Title must be changed.

[Response] Our dataset targets soil micronutrients that are essential for plant nutrition, rather than the broader category of trace elements. To avoid confusion, we have revised the terminology throughout the manuscript and changed the Title to explicitly refer to "micronutrients".

[Comment 9] Line 10: How many samples were collected in the 526 sites? In other words, please provide the size of the dataset. Moreover, were all the soils from

surface layer? How deep of the layer? At least, this basic information should be provided in the Abstract.

[Response] In this study, soils were collected from 526 sites across the Plateau, with three replicates per site, all from the surface layer (0-10 cm). We have revised the Abstract and Methods to clearly state the total sample size and sampling depth: "We assembled a plateau-wide dataset from 526 sites with triplicate surface soils (0-10 cm) per site (n = 1,660)".

**[Comment 10]** Lines 13-16: These results are too simple to summarize the characteristics of the elements, such as the concentration ranges, the reasons of the distribution, and/or potential application. Additionally, why did you only introduce the spatial patterns of Fe, Mn, and Zn, and how about the distribution of other elements? As a whole, the section of Abstract is too simple, and I cannot find more information of the dataset.

[Response] We appreciate the reviewer's valuable comment. In the revised manuscript, we have expanded the Abstract to provide more details on the dataset, including the content ranges of all six elements, major factors regulating their distributions, and potential applications.

In the original manuscript, only four elements were highlighted because the spatial prediction models for Cu and Mo performed less robustly. In the revised manuscript, we summarize the spatial patterns of all six micronutrients (Fe, Mn, Cu, Zn, Ni, V), rather than focusing only on Fe, Mn, Zn and Ni. These revisions make the Abstract more informative and better reflect the scope of the dataset.

The updated Abstract: "Soil micronutrients supply sustains critical ecological functions but exhibit poorly quantified distribution patterns in high-altitude ecosystems. This study bridges this knowledge gap through a large-scale investigation across the Tibetan Plateau, a cold-arid region where cryogenic weathering, aridity, and suppressed pedogenesis interact to govern micronutrient cycling. We assembled a plateau-wide dataset from 526 sites with triplicate surface soils (0-10 cm) per site (n = 1,660). Six micronutrients (Fe, Mn, Cu, Zn, Ni, V) were measured and paired with multi-source predictors (climate, vegetation, soil properties, topography, graze disturbance, and weathering proxy). Elemental contents span broad ranges, with site-level summaries (mean  $\pm$  SD, mg kg-1) of Fe 22,864.30  $\pm$  7,589.01, Mn 576.74  $\pm$  206.44, Cu 25.32  $\pm$  9.28, Zn 27.24  $\pm$  8.55, Ni 49.35  $\pm$  10.98, and V 56.99  $\pm$  19.33. Random Forest modeling was employed to quantify controls and generate high-resolution spatial maps. Key results reveal that pronounced regional heterogeneity driven primarily by climate-related weathering intensity and

topography variable, with secondary modulation from soil and vegetation factors. Element-specific spatial patterns were observed, with Fe enrichment in southeastern/southern plateaus, Mn gradients increasing southwestward and Zn hotspots in central-eastern and western marginal zones. Ni enriched across the northern-central interior and western highlands, Cu over the western-northern plateau with minima in the southeast, and V exhibits a moderate spatial gradient, with higher contents in the southeastern Tibetan Plateau and relatively lower values in the northwest. We provide 1-km maps of all six micronutrients together with pixel-wise uncertainty layers to support benchmarking of process-based micronutrient cycling models and to inform sustainable ecosystem management under climate change. The dataset is openly available at TPDC (https://doi.org/ 10.11888/Terre.tpdc.302870; Huo et al., 2025)".

[Comment 11] Line 20: I think this dataset only targeted to micronutrients, right? If so, the Title indeed needs to be changed to fit the contents or aims in the study.

[Response] Yes, this dataset targets soil micronutrients rather than the broader category of trace elements. To better reflect the scope and aims of the study, we have revised the Title (Mapping key soil micronutrients across the Tibetan Plateau) to refer to "micronutrients".

[Comment 12] Lines 20-22: Seriously, trace elements or micronutrients include more than those listed here. For example, BNF processes also need other trace and/or micro-elements like V, but in this dataset, many these kinds of trace elements were not considered. So, adding more elements is necessary for such dataset.

[Response] Following the reviewer's suggestion, we added vanadium (V) to the dataset and main text. Other elements such as B were not measured. We believe that the inclusion of V strengthens the dataset and improves its applicability for future studies.

[Comment 13] Line 34: Please make clear of "microelements" or "micronutrients". [Response] We have revised the manuscript and use the term micronutrients throughout the text.

[Comment 14] Lines 35-36: Add references here.

[Response] We have added references to support the statements: Current research is largely restricted to localized transects (e.g., Heihe River Basin, Tibetan Plateau Highway) with limited spatial representation (Zhang et al., 2012; Guan et al., 2017; Bu et al., 2016).

[Comment 15] Lines 40: Still, "trace elements" or "micronutrients"?

[Response] We have revised the manuscript and use the term micronutrients throughout the text.

[Comment 16] Lines 44-46: According to the figure, the farmlands were not considered in the dataset. On the Tibetan Plateau, farmland is one of the most important land uses, and more strikingly, micronutrients in farmlands are particularly essential for crops and human health, as you mentioned in Lines 23-24. Unfortunately, this dataset ignored the data in such important landscape. So, it is necessary to add the data in the farmlands for meeting the aim in this dataset (see the Title in Line 1).

[Response] Thanks for this insightful comment. On the Tibetan Plateau, cropland area is relatively small, accounting for less than 2% of the region according to the 1:1,000,000 Vegetation Map of China (Figure R6). In our original manuscript, cropland samples (n = 12, ~2% of total) were merged into the grassland category due to their small representation. As the reviewer suggested, we have treated cropland as an independent ecosystem type in our spatial comparison and up-scaling analysis (Figure R7) to provide a more accurate representation of micronutrient distributions across different ecosystem types on the Plateau.

[Comment 17] Lines 50-51: As I mentioned above, you ignored the agricultural ecosystem.

[Response] In the revised manuscript, agricultural ecosystems have been explicitly included as an ecosystem type in both descriptive and predictive analyses, as detailed in our response to Comment 4 and 16. This adjustment allows us to better account for the influence of agricultural ecosystems on soil micronutrient distributions.

[Comment 18] Lines 51-52: How did you realize "maintaining relative homogeneity in species composition, community structure, and habitat conditions"? I think this is not the necessity for the sampling in this study, because this dataset needs to represent the heterogeneity of the field on the TP. More importantly, if you had avoided these conditions, artifact disturbance must affect the analysis results of the element distribution.

[Response] In our sampling design, three replicates were collected at each site. The aim of maintaining relative homogeneity in species composition, community structure, and habitat conditions was to minimize within-site variability. This approach ensures that each site can better represent its corresponding community type, while the dataset as a whole still captures the heterogeneity across the Plateau.

[Comment 19] Lines 53-54: I have a big concern for the sampling design in this study. Clearly, the soil development is totally different in the selected ecosystems. For example, in many forests, 0-10 cm soil may only cover organic layer with high organic matter or high concentrations in some elements like Cu, Zn, or Ni, but deficiency of some other micronutrients. However, in deserts or meadows, this soil may cover the A horizon or parent materials due to the weak pedogenesis. Such a disparity will result in totally different elements' distribution in these ecosystems selected. So, the authors must provide the reasons for the sampling design in order to better direct the application of the dataset.

[Response] We appreciate the reviewer's concern regarding differences in soil profile development across ecosystems. Indeed, the 0-10 cm depth may represent the organic horizon in some forests but mainly corresponds to the mineral surface horizon in steppes, meadows and other alpine vegetation. On the Tibetan Plateau, steppes and meadows are the dominant vegetation types, together covering more than 60% of the total area, whereas forests account for less than 20%. Therefore, the majority of our samples represent mineral surface soils. We chose the 0-10 cm depth because it is the most biologically active layer for soil-plant-microbe interactions and best reflects ecological processes and nutrient cycling. This layer (0-10 cm) is widely used in regional soil surveys to ensure comparability across sites.

In the revised manuscript, we have clarified this rationale for our sampling depth in the Methods: "This depth was chosen because it represents the biologically active surface horizon most relevant to plant uptake and microbially mediated cycling, and is widely used in regional soil inventories for comparability across ecosystems. On the Tibetan Plateau, steppes and meadows constitute the dominant land cover, so most samples correspond to mineral A horizons; in forested sites, the 0-10 cm layer may include an organic-rich surface. This consistent protocol ensures cross-site comparability while capturing the variability of surface layer that most strongly interacts with vegetation and climate".

We have also added statement that differences in soil development across ecosystems may contribute to the observed variation in micronutrient distributions, which should be considered when applying the dataset: "We note that ecosystem-specific pedogenesis (e.g., organic-rich surface layers in some forests vs. thin mineral A horizons in grasslands) can contribute to differences in surface micronutrient contents. Consequently, part of the spatial variation we report likely reflects vertical horizon contrasts sampled at a uniform 0-10 cm depth. Users should consider this context when applying the dataset, especially for process inference or when comparing across ecosystems with contrasting surface horizons. Where applications

require deeper profiles or horizon-specific interpretation, we recommend integrating our maps with local profile data".

[Comment 20] Line 54: Another concern for this sampling is that elevation and vegetation community are important factors for element distribution. However, this specific information was not provided.

[Response] In the revised manuscript, we provided detailed site-level information on elevation and vegetation community in the Supplementary Table (Table R1). We also clarified in the Methods that "The sites span broad environmental gradients, ranging from 759 to 5565 m in elevation, -7.83 to 18.46 °C in mean annual temperature (MAT), and 23 to 898 mm in mean annual precipitation (MAP), effectively capturing the plateau's topographic and climatic variability".

[Comment 21] Lines 56-62: I do not think this method can well analyze the element concentrations in soil like that of ICP-OES (or ICP-AES) and ICP-MS. The XRF method has a very large error, particularly used in the field. Nowadays, this instrument is normally used in the lab, after collecting the soil samples, because it is unstable for it when using in the field. So, the authors must provide serious and strict evidence for the determination of element concentrations by using this method, and some necessary comparisons must be done with other reported data in the soil from some similar sites on the TP. Then, the quality of the data must be strictly analyzed to make sure that the element concentrations are really accurate or reasonable. At least, now I do not see the quality control data in the manuscript, and I also do not think this method could obtain reliable concentrations for most of the elements analyzed.

[Response] We sincerely thank the reviewer for this important concern. To ensure data reliability, all samples were re-analyzed in the laboratory using a third-generation XRF on air-dried, and sieved (<2 mm) samples, and the dataset presented in this study is based on these laboratory results. To further validate the measurements, 218 randomly selected samples were analyzed using ICP-MS, a widely accepted reference method. Strong correlations were observed between XRF and ICP-MS for Fe, Mn, V and Cu (R2 = 0.7-0.9), while Zn and Ni also showed significant correlations (R2 = 0.4-0.5). Taking ICP-MS as the benchmark, XRF slightly over- or underestimated absolute contents of and Ni. To address, we calibrated XRF-based Zn and Ni using the regression relationships between the two methods and repeated spatial mapping. Results show that the spatial distribution patterns before and after calibration were highly consistent, confirming that the XRF measurements are reliable for capturing spatial patterns. For Mo, the results obtained from the two methods were not fully consistent, likely due to its low content. The XRF-based measurements may involve considerable uncertainties. Therefore, we

have included both the observational and predicted results in the supplementary materials and discussed their potential uncertainties in detail in the Discussion section. We have added quality control details in the Methods, explicitly discussed methodological uncertainties in the Discussion, and provided comparative results in the Supplementary Information. Details are provided in our response to Comment 1.

[Comment 22] Line 63: In Table 1, much more information should be complemented, such as more dominated plant species, elevation ranges, local climate. I suggest to establishing more columns to exhibit this information.

[Response] Following your suggestion, we added more information in revised Table R1, including dominant plant species, elevation ranges, and local climate conditions.

**Table R1.** Ecosystem classification and sampling coverage on the Tibetan Plateau.

| Biome  | Vegetation and dominant species                                                                                                                                                                                                                                                                                                                              | Elevation   | Climate
(temperature/
precipitation) | No. of samples | No. of locations |
|--------|--------------------------------------------------------------------------------------------------------------------------------------------------------------------------------------------------------------------------------------------------------------------------------------------------------------------------------------------------------------|-------------|--------------------------------------------|----------------|------------------|
| Steppe | Alpine steppes, dominated by cold-adapted herbaceous species such as Stipa purpurea , features sparse vegetation adapted to cold-arid conditions.                                                                                                                                                                                                     | 1961-5151 m | -7.1-8.0 °C;
34-786 mm                  | 569            | 180              |
| Meadow | Alpine meadows feature dense, low-stature vegetation sustained by year-round low temperatures, high humidity, and water-retentive soils. These ecosystems thrive on gentle slopes and valley floors at higher elevations, hosting relatively diverse flora with characteristic dominance of sedges including Kobresia pygmaea and K. humilis . | 2661-5565 m | -7.8-9.2 °C;
158-849 mm                 | 499            | 154              |
| Forest | Forests on the Tibetan Plateau concentrate primarily in the southeastern region, dominated by high-altitude cold-temperate coniferous forests.  These humid-adapted ecosystems feature fir (Abies) and spruce (Picea) species as characteristic components.                                                                                                  | 1453-4237 m | -0.6-16.5 °C;
400-899 mm                | 265            | 87               |
| Shrub  | Tibetan shrublands primarily occur in arid and alpine zones, characterized by low-growing, drought-tolerant dwarf shrubs such as Lonicera (honeysuckle) and Rhododendron species adapted to nutrient-poor soils and extreme climatic conditions.                                                                                                      | 2169-5022 m | -5.2-11.8 °C;
301-869 mm                | 190            | 64               |

| Desert   | Alpine deserts occur in extremely arid, cold regions and exhibit extremely sparse vegetation dominated by arid-tolerant dwarf shrubs and herbs .                                                                                                                                                                 | 2108-5158 m | -7.1-5.6 °C;
22-443 mm  | 101 | 29 |
|----------|--------------------------------------------------------------------------------------------------------------------------------------------------------------------------------------------------------------------------------------------------------------------------------------------------------------------------------|-------------|----------------------------|-----|----|
| Cropland | Cropland, concentrated in river valleys and basin floors and is dominated by highland barley (Hordeum vulgare var. nudum, "qingke"), with spring wheat (Triticum aestivum), rapeseed (Brassica napus), and potato (Solanum tuberosum) commonly cultivated; vegetation cover is strongly seasonal and often bare after harvest. | 759-4360 m  | 0.5-18.4 °C;
113-783 mm | 36  | 12 |

Biomes are grouped by diagnostic characteristics; dominant plant species are italicized. Elevation range gives the minimum–maximum elevation (m) of sampling sites within each biome. Climate conditions report site-level ranges of mean annual temperature (°C) and mean annual precipitation (mm). No. of samples is the number of soil samples analyzed (0-10 cm, three replicates per site), and No. of locations is the number of sampling sites.

[Comment 23] Lines 70-72: This TP method is wrong, but your method is to analyze bioavailable fraction of P. Still, you must provide necessary quality control data for the precision of the element concentrations. This is particularly important for the dataset.

[Response] We apologize for the misleading statement. Phosphorus was not included in the analyses of this study, and therefore this part has been removed from the Methods section in the revised manuscript.

[Comment 24] Lines 73-76: Specify the method of CIA with necessary citation. You used XRF too much for the element analysis, but without necessary precision analysis. This is unacceptable.

**[Response]** To characterize the degree of soil development on the Plateau, we introduced the Chemical Index of Alteration (CIA), a widely used geochemical indicator of chemical weathering in soils and parent materials (Fedo et al., 1995). CIA reflects the relative loss of mobile base cations (Ca, Na, K) compared with the enrichment of immobile Al during weathering (McLennan, 1993). Higher CIA values indicate stronger chemical weathering and more advanced soil development, whereas lower values suggest weaker weathering and limited leaching of base cations (Nesbitt et al., 1982). It is calculated on a molar, anhydrous basis as CIA =  $[Al_2O_3 / (Al_2O_3 + CaO^* + Na_2O + K_2O)] \times 100$ . The information has been added into Methods section.

In addition, regarding quality control of XRF data, please refers to Response to Comment 1, where we detail our validation against ICP-MS and add quality-control information.

[Comment 25] Lines 79-80: Normally, when sampling in the field, slope, aspect, and elevation data can be recorded simultaneously. Why did you not get these data, but dependent on the online data? This will lead to more errors for them. The same case is also for the vegetation types (Lines 84-85).

[Response] We apologize for the misleading wording in the previous version. In fact, vegetation type, elevation, slope, and aspect at each sampling site were recorded simultaneously during fieldwork. For the spatial prediction of micronutrient distributions, however, the gridded predictor variables (e.g., elevation, slope, aspect, vegetation cover) were derived from online datasets in order to upscale site-level observations to the Plateau scale. We have revised the manuscript to clarify this distinction.

**[Comment 26]** Lines 88-91: Where are these data listed in your dataset, corresponding to your sites? Also, I strongly suggest the authors to providing an Excel file to exhibit all the data analyzed or compiled from online. This will help users easily obtain and cite the data.

[Response] Following your suggestion, we have provided an Excel file that lists all site-specific data, including both the field-measured variables and the compiled online predictors.

[Comment 27] Line 93: After your screening, how many data were left for the analysis below?

**[Response]** After data screening (mean  $\pm$  3 SD, by element), the retained sample sizes are: Fe 1654 (from 1660), Mn 1630 (from 1646), Cu 920 (from 946), Zn 1655 (from 1661), Ni 180 (from 181), and V 857 (from 867). We have clarified this information in the revised Methods section.

**[Comment 28]** Lines 99-100: Some nutrients like P, S were not included in this analysis? Additionally, I do not think the anthropogenic disturbance can be totally represented by grazing intensity, because in some ecosystems like deserts or forests, very little grazing activity is there. Meanwhile, this dataset did not consider the data in farmlands, which subjectively removed the important human disturbance on the TP.

[Response] We thank the reviewer for this comment. Phosphorus and sulfur were not analyzed in this study, and therefore related description has been removed from the updated manuscript.

Grazing intensity was included as a key factor because it represents the dominant form of human disturbance in the grassland ecosystems of the Tibetan Plateau (Harris et al., 2010; Yu et al., 2021). We agree that croplands are an important anthropogenic

landscape. Although croplands cover less than 2% of the Plateau, we have revised the manuscript to treat them as an independent ecosystem type in our analyses, as detailed in our response to Comment 4.

[Comment 29] Lines 109-120: Re-organize the description of the results. If you tried to introduce the distribution of element concentrations (e.g., mean, standard error), specify all the values of each element. Do not make repeated description in two different paragraphs with different aims. Additionally, please do compare your data with other reports in the similar study areas. This can help to correct the data quality in your study.

[Response] Following your suggestion, we have re-organized the description of the results to clearly present the distribution of element contents, and removed repeated descriptions as follows: "Across all sites, soil micronutrient contents varied widely, with mean value of  $22,864.30 \pm 7,589.01$  for Fe (mean  $\pm$  SD, mg kg-1),  $576.74 \pm 206.44$  for Mn,  $27.24 \pm 8.55$  for Zn,  $25.32 \pm 9.28$  for Cu,  $49.35 \pm 10.98$  for Ni and  $56.99 \pm 19.33$  for V. Coefficients of variation (CV = SD/mean) were 33% for Fe, 36% for Mn, 37% for Cu, 31% for Zn, 22% for Ni and 34% for V. Collectively, Fe and Mn dominate in absolute abundance, whereas Cu–Zn–Ni–V occur at tens of mg kg-1, indicating heterogeneous but orderly micronutrient levels across the Plateau". Our reported mean contents and variability fall well within very limited reported TP ranges, for which the possible reasons are discussed in the Discussion section (Figure R8) (also see Response to Comment 34).

[Comment 30] Line 125: ...vegetation... There are format errors in the manuscript. [Response] The formatting error has been corrected in the revised manuscript.

[Comment 31]: In Figure 3, was the statistical analysis conducted? If so, add the statistical results in the figure. The similar case is also for Figure 4.

[Response] Comment accepted. We have added the statistical results to both figures and updated results description in the revised manuscript (Figures R6 and R9).

**Figure R9.** Variability in soil micronutrient contents (Fe, Mn, Cu, Zn, Ni, V) across Tibetan lithological classes. Abbreviations of lithological classes: IA = acidic igneous rock, MA = acidic metamorphic rock, SC = clastic sedimentary rock, SO = carbonate rock, UE = eolian facies rock, UF = fluvial facies rock.

[Comment 32] Lines 125, 126 & 147: In these two sections, elemental differences among vegetation types and lithology were analyzed. However, elevation and climate gradients are also very important for the element distribution. Why not exhibit the variations in each element concentration with them? This trend is different with the analysis in the section of 3.4 (Line 175).

[Response] Thanks for this helpful suggestion. In the revised manuscript, we have added analyses of elemental variations along elevation and climate gradients (Figures R10 and R11). Results show that elevation-related variations in soil micronutrient are limited, and some elements (e.g., Cu) remain relatively stable; Fe/Mn/Zn/V show similar patterns, with higher contents at lower elevations than at higher ones. The direct relationships between trace elements and temperature or precipitation do not show clear spatial patterns, possibly due to the interactions among multiple influencing factors. Therefore, we used partial dependence plots to control for other variables to better explore the effects of climatic conditions on soil micronutrients. This part of the results has already been included in the original main text.

**Figure R10.** Variability of soil micronutrients (Fe, Mn, Cu, Zn, Ni, V) in the Tibetan Plateau with elevation gradient. Elevation classes follow the 1:1,000,000 physiognomic regionalization standard of China: mid elevation 1000-3500 m, high elevation 3500-5000 m, and very high elevation > 5000 m (Zhou et al., 2009).

**Figure R11.** Relationships between mean annual temperature (MAT) and mean annual precipitation (MAP) and the contents of six soil micronutrients (Fe, Mn, Cu, Zn, Ni, and V) across the Tibetan Plateau.

[Comment 33] Line 207: In this section, I have several concerns for the predicted results. First, because the dataset did not consider other landscapes (e.g., farmlands), the spatial patterns cannot be exactly representative on the TP. Second, even though the concentrations could be acceptable using the XRF method, I still suspect the reliability and reasonability of the data. Such a way to exhibit the concentrations of trace elements in the soil across TP may have limited reality for application. Third,

despite the statistical analysis for the prediction in Lines 208-215, there will be many uncertainties (as the authors also mentioned in Lines 255-261) for the spatial distribution of the elements, even without considering the analysis precision of the XRF method.

Overall, I really warry about the future use of the dataset under the current results, which indeed ignored the precise analysis methods by high-precision equipment such as ICP-OES, ICP-MS. Meanwhile, the AI models also have many uncertainties for the data predicted, one being also closely related to the quality of the original element concentrations.

[Response] Thanks very much for these thoughtful comments. However, we remain confident in the reliability of our extensive field measurements and the resulting spatial distribution maps derived from them. First, regarding the ecosystem coverage, although croplands account for less than 2% of the Tibetan Plateau, in the revised manuscript we have treated cropland as an independent ecosystem type in both the descriptive and predictive analyses (see our responses to Comments 4 and 16). This adjustment improves the representativeness of the dataset.

Second, concerning data reliability, as described in our response to Comment 1, all samples were re-analyzed in the laboratory using a third-generation XRF, and 218 randomly selected samples were validated against ICP-MS. Strong correlations were observed (R2 = 0.7-0.9 for Fe, Mn, Cu, and V). Zn and Ni also showed significant relationships, albeit with some systematic deviations. To address this, we calibrated XRF-based Zn and Ni using the regression relationships between the two methods. Detailed quality control results are provided in the Supplementary Information, and methodological uncertainties are explicitly discussed in the revised manuscript.

Third, regarding prediction uncertainty, we agree that model-based extrapolations involve uncertainties. In the revised Discussion, we provided uncertainty maps for all elements based on the inter-pixel variance among tree predictions (**Figure R12**) (Details please see our responses to Comments 6, #2). We further emphasize that the predicted maps are intended to provide insights into large-scale spatial patterns rather than to replace site-level high-precision measurements. We also highlight that future efforts integrating larger sample sizes and multiple analytical approaches (e.g., ICP-based methods) would further improve the robustness of such predictions. We added a subsection to discuss why Cu and Mo exhibit weaker predictive performance (also see reply to Comment 1).

In summary, while we acknowledge the limitations raised by the reviewer, we believe that the revised dataset represents the first Plateau-scale assessment of soil micronutrients with quality control and quantified uncertainties. It provides a valuable resource for understanding spatial heterogeneity and ecological drivers of micronutrient distributions across the Tibetan Plateau.

**Figure R12.** Spatial uncertainty of soil micronutrients (Fe, Mn, Cu, Zn, Ni, V) predictions across the Tibetan Plateau, expressed as standard deviation of per-tree predictions in the random forest; units in mg kg-1.

[Comment 34] Lines 228-229: This comparison is not meaningful. As I mentioned above, you should make comparisons with other reports across the study areas, and then ensure the quality of the data. Then, you may make more comparisons with other reports worldwide.

[Response] We added comparisons of our results with the few available observations on the Tibetan Plateau, as well as broader datasets from other regions, to better place our findings in a wider context (Table S2). "In terms of data range and mean values, the contents and magnitudes of the soil micronutrients we observed are generally consistent with previous reports from the Plateau and global grassland soils, indicating the reliability of our datasets. For the Plateau, the mean Mn and V contents in this study were slightly lower, while Ni were somewhat higher, and Zn was markedly lower (Figure R8). These discrepancies are more likely due to differences in study regions. Our sampling covered a much broader area, whereas previous studies focused mainly on local regions such as the Heihe Basin (Bu et al., 2016). Given the substantial spatial heterogeneity of soil trace elements across the Tibetan Plateau, such differences are expected and further highlight the necessity of exploring soil micronutrient patterns at the plateau scale".

**Table R2.** Descriptive statistics of micronutrients from various studies (in milligrams per kilogram).

|    | This study |           |           |           | China
(CNEMC,
1995) | Global UCC (Taylo (Bowen, 1979) al., 1995) |        |
|----|------------|-----------|-----------|-----------|---------------------------|--------------------------------------------|--------|
|    | Mininum    | Maxinum   | Mean      | Median    | Mean                      | Mean                                       | Mean   |
| Fe | 3,339.62   | 54,877.54 | 22,864.30 | 22,661.92 | 29,400                    | 40,000                                     | 35,000 |
| Mn | 51.05      | 1,833.82  | 576.74    | 551.01    | 583                       | 600                                        | 600    |
| Cu | 14.13      | 77.18     | 25.32     | 23.06     | 22.60                     | 30                                         | 25     |
| Zn | 7.53       | 69.19     | 27.24     | 27.12     | 74.20                     | 50                                         | 71     |
| Ni | 34.80      | 94.66     | 49.35     | 45.69     | 26.90                     | 40                                         | 20     |
| V  | 27.83      | 121.70    | 56.99     | 54.33     | 82.7                      | 90                                         | 53     |

[Comment 35] Line 230: Because of your method for analysis of element concentrations, I cannot believe the conclusion of "deficient levels" here.

[Response] We understand the reviewer's concern. As detailed in our response to Comment 1, the XRF measurements used in this study were validated against ICP-MS and showed strong correlations, confirming the reliability of the dataset. Considering that comparisons of elemental contents across regions based solely on mean values and their variability may be uncertain, we have removed this comparative analysis and the corresponding conclusion of "deficient levels".

[Comment 36] Lines 232-233: Seriously, what are the aims of this discussion or this conclusion? You did not analyze any specific fractions of elements in the soil or some other related research in the study area, and how can you conclude the increased degradation? This may mislead readers.

[Response] We agree that the original statement exceeded beyond what our data can directly support. In the revision we removed any conclusion implying "degradation/increased degradation" and reframed the text as a more cautious mechanistic discussion: "ongoing regional warming and associated hydroclimatic shifts may interact with hydrology and redox processes, organic-matter cycling, and vegetation patterns, thereby altering the availability and spatial variability of certain micronutrients (Myers et al., 2014; Pachauri et al., 2014)".

[Comment 37] Lines 236-245: Where is the direct evidence of these discussion points? I don't like discussion that lacks evidence from this study, but only based on points from cited references. As shown in your data, you have climate and weathering related data (e.g., MAP, CIA), and you should analyze these data and then make deep discussion. If the discussion is from your data, please show the relevant results in the form of figures.

[Response] Thank you for this constructive comment. Lines 236–245 of the original manuscript discuss the U-shaped response of soil micronutrients to mean annual precipitation (MAP). In the revised manuscript, we have added new analyses and figures based on our dataset to directly support this discussion.

First, we conducted a more rigorous statistical analysis using accumulated local effects (ALE) to characterize the nonlinear relationships between soil micronutrients and MAP (Figure R13). For Fe, Mn, Cu, Zn, and V, the response curves followed a decrease-stabilization-increase pattern, forming a clear U-shaped response. Both Partial Dependence Plots (PDP) and ALE analyses identified similar trends, confirming the robustness of the nonlinear precipitation effects on soil micronutrient distributions. This pattern likely reflects a trade-off between weathering, leaching losses and element retention under contrasting hydrological conditions.

We further analyzed the relationships between MAP and CIA (Chemical Index of Alteration), and found a significant positive correlation (**Figures R14 and R15**), indicating that higher precipitation generally enhances chemical weathering intensity. However, this result alone cannot fully explain the observed U-shaped micronutrient response. The available data lack direct indicators of leaching intensity and mineral retention processes. Therefore, we treat this U-shaped pattern, supported by multiple lines of evidence from our own data, as an open question for future discussion, and we acknowledge that its mechanistic explanation requires further investigation with additional datasets (e.g., redox indicators, mineral data).

**Figure R13.** Nonlinear responses of soil micronutrient contents to mean annual precipitation (MAP) estimated with accumulated local effects (ALE). The short ticks (rugs) beneath each graph indicate the distribution density of the samples along the MAP axis.

**Figure R14.** Nonlinear responses of the chemical index of alteration (CIA) to mean annual precipitation (MAP) estimated with accumulated local effects (ALE). The short ticks (rugs) beneath each graph indicate the distribution density of the samples along the MAP axis.

**Figure R15.** Responses of soil micronutrient contents to the chemical index of alteration (CIA) estimated with accumulated local effects (ALE). The short ticks (rugs) beneath each graph indicate the distribution density of the samples along the CIA axis.

[Comment 38] Lines 246-254: Similar to those in Lines 236-245, this discussion is too superficial. These discussion points are very arbitrary and lack scientific basis and evidence.

[Response] we have added quantitative analyses (Figure R16) based on our dataset to support the discussion. Grouped comparisons across aridity classes (humid, dry sub-humid, semi-arid, arid, and hyper-arid) were performed. The results show that the medians and means of Fe, Mn, Cu, Zn, Ni, and V decrease with increasing aridity, with the largest declines observed for Fe, Mn, Zn, and V. This quantitative evidence directly supports our discussion that aridity exerts a strong climatic control on soil micronutrient distributions.

**Figure R16.** Variability of soil micronutrients (Fe, Mn, Cu, Zn, Ni, V) in the Tibetan Plateau with drought gradient. Drought classes follow the Trabucco et al., 2018: Humid (AI > 0.65), Dry sub-humid (0.50  $\leq$  AI  $\leq$  0.65), Semi-arid (0.20  $\leq$  AI  $\leq$  0.50), Arid (0.03  $\leq$  AI  $\leq$  0.20), and Hyper-arid (AI  $\leq$  0.03).

In the end, we would like to express our sincere gratitude to the reviewer once again for the valuable time, effort, and constructive feedback that have greatly helped improve the quality of our manuscript.

**References:**

- Bowen H J M. Environmental chemistry of the elements, Academic Press, New York, 1979.
- Bu, J. W., Sun, Z. Y., Zhou, A. G., Xu, Y. N., Ma, R., Wei, W. H., and Liu, M.: Heavy metals in surface soils in the upper reaches of the Heihe River, Northeastern Tibetan Plateau, China, 13, https://doi.org/10.3390/ijerph13030247, 2016.
- Cheng, Y. A. and Tian, J. L.: Background values of elements in Tibetan soil and their distribution, Science Press, Beijing, 1993.
- China National Environmental Monitoring Center (CNEMC): Background Values of Soil Elements in China, China Environmental Science Press, Beijing, 1990.
- Fedo, C. M., Nesbitt, H. W., and Young, G. M.: Unraveling the effects of potassium metasomatism in sedimentary rocks and paleosols, with implications for paleoweathering conditions and provenance, Geology, 23, 921-924, https://doi.org/10.1130/0091-7613(1995)023<0921:UTEOPM>2.3.CO;2, 1995.
- Guan, Z. H., Li, X. G., and Wang, L.: Heavy metal enrichment in roadside soils in the eastern Tibetan Plateau, Environmental Science and Pollution Research, 25, 7625-7637, https://doi.org/10.1007/s11356-017-1094-8, 2017
- Harris, R. B.: Rangeland degradation on the Qinghai-Tibetan plateau: A review of the evidence of its magnitude and causes, Journal of Arid Environments, 74, 1-12, https://doi.org/10.1016/j.jaridenv.2009.06.014, 2010.
- Lemière, B.: A review of pXRF (field portable X-ray fluorescence) applications for applied geochemistry, Journal of Geochemical Exploration, 188, 350-363, https://doi.org/10.1016/j.gexplo.2018.02.006, 2018.
- McLennan, S. M.: Weathering and global denudation, The Journal of Geology, 101, 295-303, https://doi.org/10.1086/648222, 1993.
- Myers, S. S., Zanobetti, A., Kloog, I., Huybers, P., Leakey, A. D., Bloom, A. J., Carlisle, E., Dietterich, L. H., Fitzgerald, G., Hasegawa, T., Holbrook, N. M., Nelson, R. L., Ottman, M. J., Raboy, V., Sakai, H., Sartor, K. A., Schwartz, J., Seneweera, S., Tausz, M., and Usui, Y.: Increasing CO2 threatens human nutrition, Nature, 510, 139-142, https://doi.org/10.1038/nature13179, 2014.
- Nesbitt, H.W., and Young, G.M.: Early Proterozoic climates and platemotions inferred frommajor element chemistry of lutites, Nature, 299, 715-717, https://doi.org/10.1038/299715a0, 1982.
- Pachauri, R. K., Allen, M. R., Barros, V. R., Broome, J., Cramer, W., Christ, R., Church, J. A., Clarke, L., Dahe, Q., and Dasgupta, P.: Climate change 2014: Synthesis report. Contribution of Working Groups I, II and III to the Fifth Assessment Report of the Intergovernmental Panel on Climate Change, IPCC, Geneva, Switzerland, 151 pp., 2014.
- Sheng, J. J., Wang, X. P., Gong, P., Tian, L. D., and Yao, T. D.: Heavy metals of the Tibetan top soils, Level, source, spatial distribution, temporal variation and risk assessment, Environmental Science and Pollution Research, 19, 3362-3370, https://doi.org/10.1007/s11356-012-0857-5, 2012.
- Simon M. N.: Inductively Coupled Plasma Mass Spectrometry Handbook, Blackwell Publishing, Oxford, 2005.
- Tack, F.M., Verloo M.G.: Chemical speciation and fractionation in soil and sediment heavy metal analysis: a review, International Journal of Environmental Analytical Chemistry 59, 225-238. https://doi:10.1080/03067319508041330, 1995.
- Taylor, S.R., and McLennan, S.M.: The geochemical evolution of the continental crust, Reviews of Geophys, 33, 241–265, https://doi.org/10.1029/95RG00262, 1995.
- Trabucco, A. and Zomer, R. J.: Global Aridity Index and Potential Evapo-Transpiration (ET0) Climate Database v2, CGIAR Consortium for Spatial Information (CGIAR-CSI) [data set], https://cgiarcsi.community, 2018.
- Yang, A., Wang, Y. H., Hu, J., Liu, X. L., and Li, J.: Evaluation and source apportionment of heavy metals in surface soils on the Tibetan Plateau, Environmental Science, 14, 886-894, https://doi.org/10.13227/j.hjkx.201907195, 2020.
- Yu, H. L., Ding, Q. N., Meng, B. P., LV, Y. Y., Liu, C., Zhang, X. Y., Sun, Y., Li, M., and Yi, S. H.: The relative contributions of climate and grazing on the dynamics of grassland NPP and PUE on the Qinghai-Tibet Plateau, 13, 3424, https://doi.org/10.3390/rs13173424, 2021.
- Zhang, H., Wang, Z. F., Zhang, Y. L., and Hu, Z. J.: The effects of the Qinghai–Tibet railway on heavy metals enrichment in soils, Science of the Total Environment, 439, 240-280, https://doi.org/10.1016/j.scitotenv.2012.09.027, 2012.
- Zhou, C. H., Cheng, W. M., Qian, J. L., Li, B. Y., and Zhang, B. P.: Research on the Classification System of Digital Land Geomorphology of 1:1000000 in China, Journal of Geo-information Science, 11, 707-724, https://doi.org/10.3724/SP.J.1047.2009.00707, 2009.

---

## Author Comment (AC2)

**Response to Reviewer #2**

**General Assessment:**

The study addresses a significant knowledge gap in high-altitude, cold-arid ecosystems. The application of machine learning for spatial prediction is appropriate and modern. The manuscript is generally well-structured, but several aspects require clarification, strengthening, and more in-depth discussion before it can be considered for publication.

**[Response]** We appreciate the reviewer's positive comments and constructive suggestions on our manuscript, which have been carefully addressed in the revised version. We provide detailed point-by-point responses below, and we believe these revisions have substantially improved the quality and scientific rigor of the work.

**Major Comments:**

[Comment 1] There is a critical ambiguity regarding the analytical method for the core six micronutrients (Fe, Mn, Cu, Zn, Ni, Mo). The text states a portable XRF was used in the field (Lines 55-56) but later describes lab-based wavelength-dispersive XRF on pressed pellets (Lines 59-61). The accuracy and validation of field-based XRF measurements for these elements, especially at low concentrations (e.g., Mo), must be explicitly detailed. The authors should clarify the protocol, report calibration metrics (R², RMSE) against certified standards, and specify if all element data came from the same method.

[Response] Many thanks for raising this important concern. In our study, we initially used a second-generation portable XRF to conduct in situ measurements (n=130), but because this device could not reliably detect Mo, a key micronutrient of our focus, all samples were re-analyzed in the laboratory using a third-generation XRF with improved detection accuracy. All data reported in this study for core micronutrients (Fe, Mn, Cu, Zn, Ni, Mo, V) were obtained from laboratory-based wavelength-dispersive XRF (third-generation XRF) on air-dried, and sieved (<2 mm) samples.

To further validate the data quality, 218 randomly selected samples were re-analyzed using reviewer mentioned ICP-MS, a widely accepted traditional methods. Strong correlations were observed between XRF and ICP-MS for Fe, Mn, Cu, V ( $R^2 = 0.7$ -0.9; Figure R1), closely aligning with the 1:1 line. For Zn and Ni,

significant correlations were also found ( $R^2 = 0.4-0.5$ ), though with systematic deviations from the 1:1 line. Taking ICP-MS as the benchmark, XRF slightly over- or underestimated absolute contents. Importantly, these deviations do not substantially affect spatial distribution patterns central to this study for the following reasons: First, the XRF-based observed values are highly correlated with the XRF-based predicted values, and the ICP-MS-based observed values are also highly correlated with the ICP-MS-based predicted values ( $R^2 = 0.85-0.95$ ; Figure R2). Second, we calibrated XRF-based Zn, and Ni using the regression relationships between the two methods and repeated spatial mapping. Results show that the spatial distribution patterns before and after calibration were highly consistent (R2=0.8-0.9; Figure R3), confirming that the main conclusions are not sensitive to these measurement uncertainties. Third, for Mo, the results obtained from the two methods were not fully consistent, likely due to its low content. The XRF-based measurements may involve considerable uncertainties (Figure R4). Therefore, we have included both the observational and predicted results in the supplementary materials and discussed their potential uncertainties in detail in the Discussion section.

**We have added data quality control details in the Methods section as follows:**

"To ensure the reliability of soil micronutrient measurements, we adopted a two-step analytical strategy. First, in situ measurements were conducted using a second-generation portable Niton X-ray fluorescence (XRF) analyzer to obtain preliminary contents under natural field conditions. However, because this instrument could not reliably detect Mo, all soil samples were subsequently air-dried, sieved (2 mm), and re-analyzed in the laboratory using a third-generation XRF. Compared with the second-generation device, the third-generation XRF offers an extended detection range and improved accuracy (Lemière, 2018). All data presented in this study are based on the laboratory XRF measurements".

To further validate the laboratory XRF results, a subset of 218 samples was randomly selected and re-analyzed using ICP-MS, a widely accepted reference method (Simon, 2005). The two methods showed strong correlations for Fe, Mn, V and Cu ( $R^2 = 0.67-0.89$ , P

**Figure R1.** Comparison between laboratory-based third-generation XRF and ICP-MS measurements for six soil micronutrients (Fe, Mn, Cu, Zn, Ni, V).

**Figure R2.** XRF-based observed values are highly correlated with the XRF-based predicted values (a-b), and the ICP-MS-based observed values are also highly correlated with the ICP-MS-based predicted values (c-d).

**Figure R3.** Spatial correlation maps of Zn and Ni contents before and after calibration of XRF measurements against ICP-MS. The strong consistency confirming that the calibration adjusted absolute contents but did not alter the large-scale spatial patterns of these elements.

**Figure R4.** Comparison between laboratory-based third-generation XRF and ICP-MS measurements for Mo. The results obtained from the two methods were not fully consistent, likely due to its low content. The XRF-based measurements may involve considerable uncertainties. Therefore, we have included both the observational and predicted results in the supplementary materials and discussed their potential uncertainties in detail in the Discussion section.

[Comment 2] The use of relative importance metrics ('betasq') is a good start, but the analysis could be significantly strengthened. Consider using alternative methods (e.g., permutation importance from the Random Forest model itself) to cross-validate the reported driver rankings.

[Response] Following the suggestion, we complemented the relative importance analysis with Random Forest permutation importance to validate the driver rankings (Figure R5). Results showed that "Soil properties were the primary contributors to the spatial variation of micronutrient contents, followed by topographic and climatic factors, whereas vegetation and grazing disturbance had relatively minor effects. Among all predictors, the Chemical Index of Alteration (CIA), slope, and soil texture exhibited the highest importance, indicating that weathering intensity, terrain, and

soil physical structure are the dominant controls on the spatial distribution of soil micronutrients. Specifically, CIA emerged as the dominant factor for Fe, Zn, and V. Topographic factors (slope) primarily influenced Mn, while climatic variables (MAP, AI) and biological productivity (NPP) contributed more to Cu and Ni variations". This result differs from that obtained using the 'betasq' relative importance metric, which is based on the assumption of a linear or approximately linear relationship between the response and predictor variables. In contrast, the Random Forest approach can capture complex non-linear and interactive effects among environmental variables. Considering that certain environmental factors may exert non-linear influences on micronutrient distributions, such as the observed U-shaped response to precipitation, we retained the permutation importance results derived from the Random Forest model in the updated version. The model's performance also improved by 3-52%, as indicated by the increase in R² from 0.25-0.54 ('betasq') to 0.39-0.77 ('Random Forest model').

**Figure R5.** Relative importance of biotic and abiotic factors for soil micronutrients (Fe, Mn, Cu, Zn, Ni, V) on the Tibetan Plateau.

[Comment 3] Furthermore, the discussion of the U-shaped response to MAP (Line 191) is intriguing but remains qualitative. A more rigorous statistical exploration of

these nonlinear relationships (e.g., using generalized additive models) would greatly bolster this key finding.

[Response] In the revision, we conducted a more rigorous statistical analysis using accumulated local effects (ALE) to characterize the nonlinear relationships between soil micronutrients and MAP (Figure R6). For Fe, Mn, Cu, Zn, and V, the response curves followed a decrease-stabilization-increase pattern, forming a clear U-shaped response. Predicted micronutrients decreased sharply under low to moderate precipitation, remained relatively stable across intermediate precipitation, and then increased again under high precipitation. Ni showed a hump-shaped pattern, with high values at low MAP followed by a gradual decline and subsequent stabilization. Both PDP and ALE analyses identified similar trends, confirming the robustness of the nonlinear precipitation effects on soil micronutrient distributions. This pattern likely reflects a trade-off between weathering, leaching losses and element retention under contrasting hydrological conditions. We have added these points into updated Discussion.

Here we chose ALE instead of GAMs because ALE captures conditional effects consistent with the Random Forest structure, avoiding biases from correlated or interacting predictors and better reflecting precipitation's nonlinear influence on micronutrients. Thanks for your understanding.

**Figure R6.** Nonlinear responses of soil micronutrient contents to mean annual precipitation (MAP) estimated with accumulated local effects (ALE). The short ticks (rugs) beneath each graph indicate the distribution density of the samples along the MAP axis.

[Comment 4] The poor performance of the models for Cu and Mo needs a more thorough discussion. Simply stating limited utility is insufficient. The authors should hypothesize why these elements are less predictable. Are the key drivers not captured in the predictor set? Is measurement error higher? Is their distribution more stochastic? This critical reflection is essential for a balanced interpretation of the results.

[Response] Thanks for the kind comment. In the revision, we added a subsection to discuss why Cu and Mo exhibit weaker predictive performance. First, analytical limitations may introduce measurement error, as both elements occur at very low contents. Second, key process controls are not fully captured by the current predictors. E.g. Cu is strongly influenced by nonlinear interactions with Fe/Al oxides, whereas Mo is affected by carbonate content, redox conditions (Tack et al., 1995). These mineralogical and redox variables are underrepresented, leading to potential bias. In addition, both elements show hotspot-prone, right-skewed distributions (Figure 2), likely due to localized anthropogenic sources. The spatial heterogeneity introduces stochastic variability beyond the range of our covariates. We have added these points into revised Discussion. Also, we added pixel-level uncertainty layers with the maps for users to properly utilize these datasets.

[Comment 5] The results show significant lithological control for some elements (Fig. 4), yet climate is reported as the dominant driver in the importance analysis (Fig. 5). This apparent discrepancy needs reconciliation. The discussion should integrate these findings, explaining how regional climate patterns might override or interact with the inherent geochemical signal from the parent material across the vast plateau.

[Response] Many thanks for this thoughtful comment. A closer examination of Fig. 4 shows that Fe, Mn, Zn, Ni, and V display slightly higher contents over acidic metamorphic rocks (MA), whereas Cu show no significant lithological differences (Figure R7). Although lithological differences are detectable for some elements, they do not alter the overall spatial patterns. The weak lithological signal for Cu further suggests limited parent-material control. In summary, lithology modulates the baseline levels of certain micronutrients but is not the dominant factor shaping their regional distributions, which are more strongly governed by external environmental drivers such as climate, weathering intensity, sediment transport, and biotic cycling. We have incorporated these points into the revised Discussion.

**Figure R7.** Variability in soil micronutrient contents (Fe, Mn, Cu, Zn, Ni, V) across Tibetan lithological classes. Abbreviations of lithological classes: IA = acidic igneous rock, MA = acidic metamorphic rock, SC = clastic sedimentary rock, SO = carbonate rock, UE = eolian facies rock, UF = fluvial facies rock.

[Comment 6] The high-resolution prediction maps (Fig. 7) are a key output. However, the manuscript does not provide associated uncertainty maps (e.g., prediction intervals). For users to properly utilize these datasets, an assessment and visualization of spatial uncertainty are crucial. Please add this or explicitly state it as a limitation. [Response] Following your suggestion, we considered the model-based uncertainty from the Random Forest algorithm and added a pixel-level uncertainty layer (Figure R8) alongside the spatial prediction maps and supplemented the calculation method in the Methods section: "The uncertainty layer is calculated as the inter-pixel variance among tree predictions".

In addition, we discussed the input-data-related uncertainties not quantified in this study. The gridded climate data were derived based on a limited number of meteorological stations, and sparse coverage at high elevations may introduce biases in the western Plateau. Likewise, the soil property data, though based on all available field observations, remain limited in remote areas, potentially causing systematic bias. These data constraints still affect prediction accuracy of soil micronutrients. Accordingly, we added the following statement to the revised manuscript: "The spatial predictions inevitably contain uncertainties arising from both the Random Forest model and the input datasets. Model-based uncertainty was quantified as the inter-tree variance among predictions, while additional uncertainty may stem from sparse meteorological observations and limited soil sampling across the Tibetan Plateau".

**Figure R8.** Spatial uncertainty of soil micronutrients (Fe, Mn, Cu, Zn, Ni, V) predictions across the Tibetan Plateau, expressed as standard deviation of per-tree predictions in the random forest; units in mg kg-1.

**Minor Comments:**

[Comment 7] The abstract mentions six microelements but the title and data availability specify only four (Fe, Mn, Zn, Ni). The title and abstract should be aligned. Either adjust the title to reflect the full study or refocus the abstract on the four well-predicted elements.

[Response] We have updated the title as follows: "Mapping key soil micronutrients across the Tibetan Plateau".

**[Comment 8]** Figure 1b is described but not effectively explained in the caption. The relationship between the bars and dots (ecosystem area vs. sampling frequency) should be explicitly stated to justify the representativeness of the sampling strategy.

[Response] We have revised the caption of Fig. 1b to state that the bars represent the areal proportion of each ecosystem, while the dots represent the proportion of sampling sites within the same ecosystem. Similar bar and dot heights indicate that the sampling is proportionally representative.

[Comment 9] The terms "micronutrients," "microelements," and "trace elements" are used interchangeably. For consistency and precision, authors should choose the same word throughout the manuscript.

[Response] To ensure consistency and precision, we use "micronutrients" throughout the manuscript.

[Comment 10] The data availability section provides a DOI, but this should also be formally cited in the main text (e.g., in the Methods or Results section) when the dataset is first mentioned.

[Response] We have added a formal citation with the DOI at the first mention of the dataset in the Abstract section.

[Comment 11] Line 40 has a trailing comma after "Mo" ("...Ni, Mo,)."). [Response] The redundant comma has been deleted.

[Comment 12] The discussion on ecological implications (Lines 228-233) is good but could be slightly expanded. Briefly mention specific plateau processes that might be most sensitive to these micronutrient limitations.

[Response] Thank you for the suggestion. Multiple biogeochemical processes are sensitive to micronutrient availability. Biological nitrogen fixation relies on Mo and Fe as essential cofactors of nitrogenase, while the methane cycle depends on Ni and Cu through their roles in methanogenesis and methane oxidation, respectively (Thauer et al., 2019; Stefan et al., 2020). Permafrost thaw and thermokarst development can further activate Fe-Mn redox cycling, altering metal mobility (Chauhan et al., 2024). Collectively, soil micronutrients depletion may cascade through nutrient cycling and ecosystem feedbacks, amplifying the impacts of ongoing environmental change across the Plateau. We have incorporated these points into revised Discussion.

In the end, we would like to express our sincere gratitude to the reviewer once again for the valuable time, effort, and constructive feedback that have greatly helped improve the quality of our manuscript.

**References:**

- Chauhan, A., Patzner, M. S., Bhattacharyya, A., Borch, T., Fischer, S., Obst, M., Thomasarrigo, L.K., Kretzschmar, R., Mansor, M., Bryce, C., Kappler, A., and Joshi, P.: Interactions between iron and carbon in permafrost thaw ponds, Science of the Total Environment, 946, 174321, https://doi.org/10.1016/j.scitotenv.2024.174321, 2024.
- Lemière, B.: A review of pXRF (field portable X-ray fluorescence) applications for applied geochemistry, Journal of Geochemical Exploration, 188, 350-363, https://doi.org/10.1016/j.gexplo.2018.02.006, 2018.
- Simon M. N.: Inductively Coupled Plasma Mass Spectrometry Handbook, Blackwell Publishing, Oxford, 2005.
- Stefan, B., Jiménez-Vicente, E., Echavarri-Erasun, C., and Rubio, L. M.: Biosynthesis of Nitrogenase Cofactors, Chemical Reviews, 120, 4919-5794, https://doi.org/10.1021/acs.chemrev.9b00489, 2020.
- Tack, F.M., Verloo M.G.: Chemical speciation and fractionation in soil and sediment heavy metal analysis: a review, International Journal of Environmental Analytical Chemistry 59, 225-238. https://doi:10.1080/03067319508041330, 1995.
- Thauer, P. K.: Methyl (Alkyl)-Coenzyme M Reductases: Nickel F-430-Containing Enzymes Involved in Anaerobic Methane Formation and in Anaerobic Oxidation of Methane or of Short Chain Alkanes, Biochemistry, 58, 5198-5220. https://doi:10.1021/acs.biochem.9b00164, 2019.